# CMamba: Channel Correlation Enhanced State Space Models for Multivariate Time Series Forecasting

## Abstract

Recent advancements in multivariate time series forecasting have been propelled by Linear-based, Transformer-based, and Convolution-based models, with Transformer-based architectures gaining prominence for their efficacy in temporal and cross-channel mixing. More recently, Mamba, a state space model, has emerged with robust sequence and feature mixing capabilities. However, the suitability of the vanilla Mamba design for time series forecasting remains an open question, particularly due to its inadequate handling of cross-channel dependencies. Capturing cross-channel dependencies is critical in enhancing the performance of multivariate time series prediction. Recent findings show that self-attention excels in capturing cross-channel dependencies, whereas other simpler mechanisms, such as MLP, may degrade model performance. This is counterintuitive, as MLP, being a learnable architecture, should theoretically capture both correlations and irrelevances, potentially leading to neutral or improved performance. Diving into the self-attention mechanism, we attribute the observed degradation in MLP performance to its lack of data dependence and global receptive field, which result in MLP's lack of generalization ability. Considering the powerful sequence modeling capabilities of Mamba and the high efficiency of MLP, the combination of the two is an effective strategy for solving multivariate time series prediction. Based on the above insights, we introduce a refined Mamba variant tailored for time series forecasting. Our proposed model, **CMamba**, incorporates a modified Mamba (M-Mamba) module for temporal dependencies modeling, a global data-dependent MLP (GDD-MLP) to effectively capture cross-channel dependencies, and a Channel Mixup mechanism to mitigate overfitting. Comprehensive experiments conducted on seven real-world datasets demonstrate the efficacy of our model in improving forecasting performance.

## 1 Introduction

Multivariate (or Multichannel) time series forecasting (MTSF) plays a crucial role in diverse applications, such as weather prediction (Chen et al., 2023), traffic management (Liu et al., 2023b;c), economics (Xu & Cohen, 2018), and event prediction (Xue et al., 2023). MTSF aims to predict future values of temporal variations based on historical observations. Given its practical importance, numerous deep learning models have been developed in recent years, notably including, Linear-based (Zeng et al., 2023; Li et al., 2023; Das et al., 2023; Wang et al., 2023), Transformer-based (Zhou et al., 2022; Zhang & Yan, 2022; Wu et al., 2021; Nie et al., 2022; Liu et al., 2023a), and Convolution-based (Liu et al., 2022; Wang et al., 2022; Wu et al., 2022; Luo & Wang, 2024) models, which have demonstrated significant advancements.

Among these, Transformer-based models are particularly distinguished by their capacity to separately mix temporal and channel embeddings using mechanisms such as MLP or self-attention. The recently introduced Mamba model (Gu & Dao, 2023), which operates within a state space framework, exhibits notable sequence and feature mixing capabilities. Mamba has shown progress not only in natural language processing but also in other domains, including computer vision (Liu et al., 2024; Zhu et al., 2024) and graph-based applications (Wang et al., 2024a). Nevertheless, recent studies on Mamba in time series (Patro & Agneeswaran, 2024; Wang et al., 2024d) mostly focus on

how to align the input format of time series with other fields so that the original Mamba architecture can be migrated while ignoring the exploration of whether the components of Mamba itself are suitable for time series. In addition, the vanilla Mamba lacks cross-channel dependency modeling capabilities. S-Mamba (Wang et al., 2024d) solves this by treating different channels as a sequence and modeling their dependencies with state space mechanisms. However, the strategy of treating channels as sequences is not practical and also results in large complexity.

Capturing cross-channel dependencies is critical in enhancing the performance of multivariate time series prediction. Recent findings (Liu et al., 2023a) show that self-attention excels in capturing cross-channel dependencies. The resulting model, iTransformer, achieves excellent performance on multiple datasets. Whereas, as another important component of the Transformer encoder block, modeling cross-channel dependencies with MLP (i.e., the FFN layer) degrades model performance (Liu et al., 2023a; Nie et al., 2022). Being a lightweight and learnable architecture, MLP should be an ideal structure to capture cross-channel dependencies as it could theoretically capture both correlations and irrelevances, leading to neutral or improved performance.

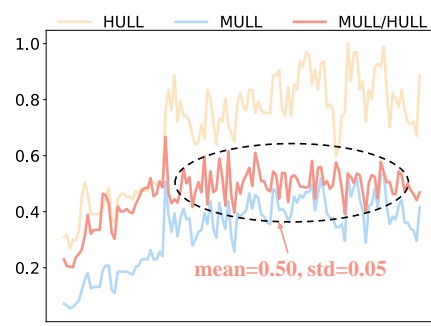

Figure 1: An illustration of the relationship of variables in the ETT dataset. *HULL* means High UseLess Load and *MULL* means Middle UseLess Load.

In the design of iTransformer, cross-channel dependencies are captured in a data-dependent and global manner. That is, each channel is characterized by its historical sequence, and the self-attention mechanism adaptively captures the dependencies between channels according to the data. In contrast, the parameters of channel mixing are the same for different inputs in the original MLP. Moreover, because the data at each time point is mixed independently during training, the model focuses too much on the local dependencies between channels while ignoring the global dependencies. The lack of data dependence and global receptive field, which are crucial for the complex, long-term dependencies observed in real-world data, results in the discrepancy between MLP and self-attention. To validate our claims, we depict the curves of two variables (channels) over time in the ETT dataset in Fig. 1. We find that: (i) Despite the fluctuations, the relationship between these two variables remains stable over a long time, i.e., *MULL* (Middle UseLess Load) is roughly equivalent to half of *HULL* (High UseLess Load). (ii) This relationship can vary across different time periods. The above observations indicate that multivariate time series have relatively stable long-term channel dependencies at different periods. Hence, a data-dependent and global channel modeling approach is more suitable for capturing cross-channel dependencies.

Based on the above motivations, we propose enhancements to Mamba for multivariate time series prediction, incorporating a modified Mamba (M-Mamba) module for long-term temporal dependencies modeling and a global data-dependent MLP (GDD-MLP) to model channel dependencies more efficiently. GDD-MLP endows the original MLP with the advantages of data dependence and a global receptive field. Additionally, to mitigate the overfitting and generalization issues associated with Channel-Dependent (CD) models (Han et al., 2023), we introduce a Channel Mixup strategy. This method combines channels linearly during training to create virtual channels that integrate characteristics from multiple channels while preserving their shared temporal dependencies, which is expected to improve the generalizability of models.

In summary, we adapt Mamba to multivariate time series forecasting by evaluating the effectiveness of its components and equipping it with channel mixing capabilities. The resulting model, **CMamba**, leveraging both cross-time and cross-channel dependencies, achieves consistent state-of-the-art performance across seven real-world datasets. Technically, our main contributions are summarized as follows:

- We tailor the vanilla Mamba module for long-term time series forecasting. A modified Mamba (M-Mamba) module is proposed for better cross-time dependency modeling.
- We enable Mamba to capture multivariate correlations using the proposed global data-dependent MLP (GDD-MLP) and Channel Mixup. Coupled with the M-Mamba module, the proposed **CMamba** captures both cross-time and cross-channel dependencies to learn better representations for multivariate time series forecasting.

- Experiments on seven real-world benchmarks demonstrate that our proposed framework achieves superior performance. We extensively apply the proposed GDD-MLP and Channel Mixup to other models. The improved forecasting performance indicates the broad versatility of our framework.

## 2 RELATED WORK

### 2.1 STATE SPACE MODELS FOR SEQUENCE MODELING

Traditional state space models (SSMs), such as hidden Markov models and recurrent neural networks (RNNs), process sequences by storing messages in their hidden states and using these states along with the current input to update the output. This recurrent mechanism limits their training efficiency and leads to problems like vanishing and exploding gradients (Hochreiter & Schmidhuber, 1997). Recently, several SSMs with linear-time complexity have been proposed, including S4 (Gu et al., 2021), H3 (Fu et al., 2022), SpaceTime (Zhang et al., 2023), and Mamba (Gu & Dao, 2023). Among these, Mamba further enhances S4 by introducing a data-dependent selection mechanism that balances short-term and long-term dependencies. Mamba has demonstrated powerful long-sequence modeling capabilities and has been successfully extended to the visual (Liu et al., 2024; Zhu et al., 2024) and graph domains (Wang et al., 2024a).

In the time series domain, many recent works (Patro & Agneeswaran, 2024; Wang et al., 2024d) have focused on aligning the input format of time series with other fields so that the original Mamba architecture can be migrated. Specifically, S-Mamba (Wang et al., 2024d) modifies iTransformer (Liu et al., 2023a) by replacing its self-attention layer with Mamba, thereby modeling inter-channel relationships by treating different channels as sequential data. Time-SSM (Hu et al., 2024) provides an in-depth examination of various SSM kernel variants within the Mamba framework for time series analysis. Bi-Mamba+ (Liang et al., 2024) explores the impact of sequence modeling direction and proposes a bidirectional Mamba architecture for multivariate time series forecasting. Despite these advancements, existing works overlook the critical evaluation of whether the internal components of Mamba are inherently well-suited for time series data. Moreover, while addressing Mamba's limited capacity for channel modeling, existing strategies are too direct (e.g., S-Mamba, which models cross-channel dependencies in a sequential manner) and introduce large computational overhead. In this paper, we explore the practicality of Mamba components and propose a more lightweight and efficient strategy to model cross-channel dependencies.

### 2.2 CHANNEL STRATEGIES IN MTSF

Channel strategies are fundamental in determining how to handle relationships between variables in multivariate time series forecasting (MTSF). Broadly, there are two primary approaches: the Channel-Independent (CI) strategy, which disregards cross-channel dependencies, and the Channel-Dependent (CD) strategy, which integrates channels according to specific mechanisms. Each strategy has its respective strengths and weaknesses. CD methods offer greater representational capacity but tend to be less robust when confronted with distributional shifts in time series data, while CI methods sacrifice capacity for more stable, robust predictions (Han et al., 2023).

A significant number of state-of-the-art models adhere to the CI strategy. These models (Zeng et al., 2023; Nie et al., 2022; Wang et al., 2023) treat multivariate time series as a collection of independent univariate time series and simply treat different channels as different training samples. However, recent work has demonstrated the efficacy of explicitly capturing multivariate correlations through mechanisms such as self-attention (Liu et al., 2023a) and convolution (Luo & Wang, 2024). These methods have achieved strong empirical results, underscoring the importance of cross-channel dependency modeling in MTSF. Despite these advances, there remains a need for more effective and efficient mechanisms to capture and model cross-channel dependencies.

### 2.3 MIXUP FOR GENERALIZABILITY

Mixup is an effective data augmentation technique widely used in vision (Zhang et al., 2017; Yun et al., 2019; Verma et al., 2019), natural language processing (Guo et al., 2019; Sun et al., 2020), and more recently, time series analysis (Zhou et al., 2023; Ansari et al., 2024). The vanilla mixup

technique randomly mixes two input data samples via linear interpolation. Its variants extend this by mixing either input samples or hidden embeddings to gain better generalization. In multivariate time series, each sample contains multiple time series. Hence, rather than mixing two samples, our proposed Channel Mixup mixes the time series of the same sample. This strategy not only enhances the generalization of models but also facilitates the CD approach.

# 3 PRELIMINARY

## 3.1 MULTIVARIATE TIME SERIES FORECASTING

In multivariate time series forecasting, we are given a historical time series $\mathbf{X} = \{\mathbf{x}_1, ..., \mathbf{x}_L\} \in \mathbb{R}^{L \times V}$ with a look-back window $L$ and the number of channels $V$. The objective is to predict the $T$ future values $\mathbf{Y} = \{\mathbf{x}_{L+1}, ..., \mathbf{x}_{L+T}\} \in \mathbb{R}^{T \times V}$. In the following sections, we denote $\mathbf{X}_{t,:}$ as the value of all channels at time step $t$, and $\mathbf{X}_{:,v}$ as the entire sequence of the channel indexed by $v$. The same annotation is also applied to $\mathbf{Y}$. In this paper, we focus on the long-term time series forecasting task, where the prediction length is greater than or equal to 96.

## 3.2 MAMBA

Given an input $\overline{\mathbf{x}}(t) \in \mathbb{R}$, the continuous state space model (SSM) produces a response $\overline{\mathbf{y}}(t) \in \mathbb{R}$ based on the observation of hidden state $\mathbf{h}(t) \in \mathbb{R}^S$ and the input $\overline{\mathbf{x}}(t)$, which can be formulated as:

$$
\begin{aligned}
\mathbf{h}'(t) &= \mathbf{A}\mathbf{h}(t) + \mathbf{B}\overline{\mathbf{x}}(t), \\
\overline{\mathbf{y}}(t) &= \mathbf{C}\mathbf{h}(t) + \mathbf{D} \cdot \overline{\mathbf{x}}(t),
\end{aligned}
\tag{1}
$$

where $\mathbf{A} \in \mathbb{R}^{S \times S}$ is the state transition matrix, $\mathbf{B} \in \mathbb{R}^{S \times 1}$ and $\mathbf{C} \in \mathbb{R}^{1 \times S}$ are projection matrices, and $\mathbf{D} \in \mathbb{R}$ is the skip connection parameter. When both input and response contain $E$ features, i.e., $\overline{\mathbf{x}}(t) \in \mathbb{R}^E$ and $\overline{\mathbf{y}}(t) \in \mathbb{R}^E$, the SSM is applied independently to each feature, that is, $\mathbf{A} \in \mathbb{R}^{E \times S \times S}$, $\mathbf{B} \in \mathbb{R}^{E \times S}$, $\mathbf{C} \in \mathbb{R}^{E \times S}$, and $\mathbf{D} \in \mathbb{R}^E$. For efficient memory utilization, $\mathbf{A}$ can be compressed to $\mathbb{R}^{E \times S}$. Hereafter, unless otherwise stated, we only consider multi-feature systems and the compressed form of $\mathbf{A}$. For the discrete system, Eq. 1 could be discretized as:

$$
\begin{aligned}
\overline{\mathbf{A}} &= \exp(\Delta\mathbf{A}), \\
\overline{\mathbf{B}} &= (\Delta\mathbf{A})^{-1}(\exp(\Delta\mathbf{A}) - \mathbf{I})\Delta\mathbf{B}, \\
\mathbf{h}_t &= \overline{\mathbf{A}}\mathbf{h}_{t-1} + \overline{\mathbf{B}}\overline{\mathbf{x}}_t, \\
\overline{\mathbf{y}}_t &= \mathbf{C}\mathbf{h}_t + \mathbf{D} \cdot \overline{\mathbf{x}}_t,
\end{aligned}
\tag{2}
$$

where $\Delta \in \mathbb{R}^E$ is the sampling time interval. These operations can be efficiently computed through global convolution:

$$
\begin{aligned}
\overline{\mathbf{K}} &= (\mathbf{C}\overline{\mathbf{B}}, \mathbf{C}\overline{\mathbf{A}}\overline{\mathbf{B}}, ..., \mathbf{C}\overline{\mathbf{A}}^{L-1}\overline{\mathbf{B}}), \\
\overline{\mathbf{Y}} &= \overline{\mathbf{X}} * \overline{\mathbf{K}} + \mathbf{D} \cdot \overline{\mathbf{X}},
\end{aligned}
\tag{3}
$$

where $L$ is the sequence length, and $\overline{\mathbf{X}}, \overline{\mathbf{Y}} \in \mathbb{R}^{L \times E}$.

**Selective Scan Mechanism** Traditional approaches (Gu et al., 2021) keep transfer parameters (e.g., $\mathbf{B}$ and $\mathbf{C}$) unchanged during sequence processing, ignoring their relationships with the input. Mamba (Gu & Dao, 2023) adopts a selective scan strategy where $\mathbf{B} \in \mathbb{R}^{L \times S}$, $\mathbf{C} \in \mathbb{R}^{L \times S}$, and $\Delta \in \mathbb{R}^{L \times E}$ are dynamically derived from the input $\overline{\mathbf{X}} \in \mathbb{R}^{L \times E}$. As a result, $\overline{\mathbf{A}}$ and $\overline{\mathbf{B}} \in \mathbb{R}^{L \times E \times S}$ are dependent on both time steps and features. This data-dependent mechanism allows Mamba to incorporate contextual information and selectively execute state transitions.

# 4 METHODOLOGY

The overall structure of CMamba is illustrated in Fig. 2. Before training, the Channel Mixup module mixes the input multivariate time series along the channel dimension. This is followed by the core architecture comprising the modified Mamba (M-Mamba) module and the global data-dependent MLP (GDD-MLP) module, designed to capture both cross-time and cross-channel dependencies.

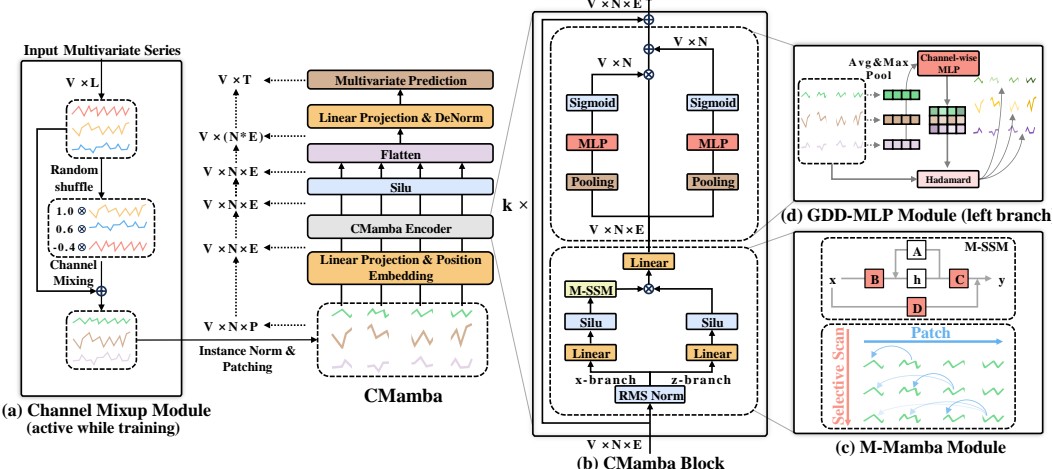

Figure 2: The overall framework of CMamba. (a) The Channel Mixup module, active during training, fuses different channels of a sample to create a virtual sample. New samples are normalized via instance norm and segmented into patches before being fed into the model. (b) The CMamba block consists of two parts: the M-Mamba module and GDD-MLP before residual connection. (c) The M-Mamba module captures cross-time dependencies. (d) The GDD-MLP module captures cross-channel dependencies.

These two modules form the CMamba block together. CMamba takes patch-wise sequences as input and makes predictions using a single linear layer. The following sections provide detailed explanations of these components.

## 4.1 CMAMBA BLOCK

The CMamba block consists of two key components: the M-Mamba module and the GDD-MLP module, each responsible for modeling cross-time and cross-channel dependencies, respectively.

### 4.1.1 M-MAMBA

Mamba has demonstrated significant potential in NLP (Gu & Dao, 2023), CV (Zhu et al., 2024; Liu et al., 2024), and stock prediction (Shi, 2024). In these fields, the semantic consistency of features allows elements like words, image patches, or financial indicators to be treated as tokens. In contrast, in multivariate time series, different channels often represent disparate physical quantities (Liu et al., 2023a), making it unsuitable to treat channels at the same time point as a token. While a single time step of each channel lacks semantic meaning, patching (Dosovitskiy et al., 2020; Nie et al., 2022) aggregates time points into subseries-level patches, enriching the semantic information and local receptive fields of tokens. Hence, we divide the input time series into patches to serve as the input of our model.

**Patching** Given multivariate time series $\mathbf{X}$, for each univariate series $\mathbf{X}_{:v} \in \mathbb{R}^L$, we segment it into patches through moving window with patch length $P$ and stride $S$:

$$\hat{\mathbf{X}}_{:v} = \text{Patching}(\mathbf{X}_{:v}), \tag{4}$$

where $\hat{\mathbf{X}}_{:v} \in \mathbb{R}^{N \times P}$ is a sequence of patches and $N = \lfloor \frac{(L-P)}{S} \rfloor + 2$ is the number of patches.

In addition to modifying the input structure, the model architecture of Mamba also needs to be adapted to suit the characteristics of multivariate time series data. Our Mamba variant is called M-Mamba, as shown in Fig. 3 (b). Compared with the vanilla Mamba module, our module has the following differences: (1) We remove the x-branch convolution operation. (2) We no longer learn a transition matrix $\mathbf{A}$ for each feature, i.e., $\mathbf{A} \in \mathbb{R}^S$ in M-Mamba. (3) The skip connection matrix $\mathbf{D}$ is derived from the input time series, meaning that $\mathbf{D}$ is also data-dependent. The reasons for our design will be discussed in detail in Section 5.2.

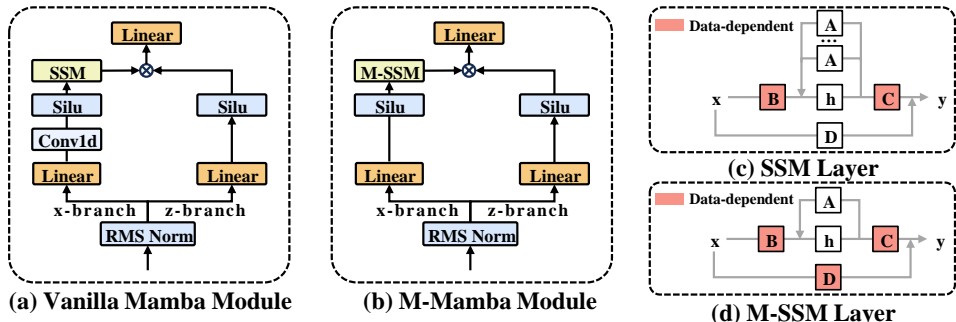

Figure 3: Architectures of the vanilla Mamba module and M-Mamba module.

### 4.1.2 GDD-MLP

To model cross-channel dependencies as simply and effectively as possible, we propose the global data-dependent MLP (GDD-MLP) module. Fig. 2 (b) and (d) illustrate its structure and pipeline. For the patch-wise multivariate time series embeddings generated after the $l^{th}$ M-Mamba module, denoted as $\mathbf{H}_l \in \mathbb{R}^{V \times N \times E}$, the data-dependent weight and bias of GDD-MLP could be formulated as:

$$\begin{aligned} \mathbf{Weight}_l &= \text{sigmoid}(\text{MLP}_1(\text{Pooling}(\mathbf{H}_l))), \\ \mathbf{Bias}_l &= \text{sigmoid}(\text{MLP}_2(\text{Pooling}(\mathbf{H}_l))). \end{aligned} \quad (5)$$

Here, Pooling is applied along the embedding dimension, generating descriptors $\mathbf{F}^l \in \mathbb{R}^{V \times N}$ that encapsulate the overall characteristics of each patch for every channel. To improve the extraction of global information for each channel, we employ both Average Pooling and Max Pooling. The average and max descriptors share the same MLP, whose outputs are added up to produce the final channel representations in the latent space. The weight and bias are then applied to the input features to embed cross-channel dependencies:

$$\mathbf{H}'_l = \mathbf{Weight}_l \odot \mathbf{H}_l + \mathbf{Bias}_l. \quad (6)$$

### 4.2 CHANNEL MIXUP

Previous mixup strategies (Zhang et al., 2017) generate new training samples through linear interpolation between two existing samples. For two feature-target pairs $(x_i, y_i)$ and $(x_j, y_j)$ randomly drawn from the training set, the process is described as:

$$\begin{aligned} \tilde{x} &= \lambda x_i + (1 - \lambda)x_j, \\ \tilde{y} &= \lambda y_i + (1 - \lambda)y_j, \end{aligned} \quad (7)$$

where $(\tilde{x}, \tilde{y})$ is the synthesized virtual sample, and $\lambda \in [0, 1]$ is a mixing coefficient. In the context of multivariate time series, directly applying this mixup approach can yield suboptimal results and even degrade model performance (Zhou et al., 2023), as mixing samples from distinct time intervals may disrupt temporal features such as periodicity. In contrast, different channels within a multivariate time series often exhibit similar temporal characteristics, which explains the effectiveness of the CI strategy (Han et al., 2023). Mixing different channels could introduce new variables while preserving their shared temporal features. Considering that the CD strategy tends to cause overfitting due to its lack of robustness to distributionally drifted time series (Han et al., 2023), training with unseen channels should mitigate this issue. Generally, the Channel Mixup could be formulated as follows:

$$\begin{aligned} \mathbf{X}' &= \mathbf{X}_{:,i} + \lambda \mathbf{X}_{:,j}, \ i, j = 0, ..., V - 1, \\ \mathbf{Y}' &= \mathbf{Y}_{:,i} + \lambda \mathbf{Y}_{:,j}, \ i, j = 0, ..., V - 1, \end{aligned} \quad (8)$$

where $\mathbf{X}' \in \mathbb{R}^{L \times 1}$ and $\mathbf{Y}' \in \mathbb{R}^{T \times 1}$ are hybrid channels resulting from the linear combination of channel $i$ and channel $j$. $\lambda \sim N(0, \sigma^2)$ is the linear combination coefficient with $\sigma$ as the standard deviation. We use a normal distribution with a mean of 0, ensuring that the overall characteristics of each channel remain unchanged. In practice, as shown in Alg. 1, we mix the channels of each sample and replace the original sample with the constructed virtual sample, where randperm($V$) generates a randomly arranged array of $0 \sim V - 1$.

---

**Algorithm 1:** Channel Mixup for multivariate time series forecasting

---

**Input:** training data $\mathbf{X} \in \mathbb{R}^{L \times V}$, $\mathbf{Y} \in \mathbb{R}^{T \times V}$; standard deviation $\sigma$; the number of channels $V$

1: perm = randperm($V$)    # perm$\in \mathbb{R}^V$
2: $\lambda$ = normal(mean = 0, std = $\sigma$, size = ($V$,))
3: $\mathbf{X}' = \mathbf{X} + \lambda * \mathbf{X}[:, \text{perm}]$
4: $\mathbf{Y}' = \mathbf{Y} + \lambda * \mathbf{Y}[:, \text{perm}]$

**Output:** $(\mathbf{X}', \mathbf{Y}')$

---

### 4.3 OVERALL PIPELINE

In this section, we summarize the aforementioned procedures and outline the process of training and testing our model.

In the training stage, given a sample $\{\mathbf{X}, \mathbf{Y}\}$, it is first transformed into a virtual sample using the Channel Mixup technique, followed by instance normalization that mitigates the distributional shifts:

$$\mathbf{X}', \mathbf{Y}' = \text{ChannelMixup}(\mathbf{X}, \mathbf{Y}),$$
$$\mathbf{X}'_{norm} = \text{InstanceNorm}(\mathbf{X}'). \tag{9}$$

Subsequently, each channel is partitioned into patches of equal patch length $P$ and patch number $N$. These patch-wise tokens are then linearly projected into vectors of size $E$ followed by the addition of a position encoding $\mathbf{W}_{pos}$. This process could be described as:

$$\hat{\mathbf{X}} = \text{Patching}(\mathbf{X}'_{norm}),$$
$$\mathbf{Z}_0 = \hat{\mathbf{X}}\mathbf{W}_p + \mathbf{W}_{pos}, \tag{10}$$

where $\hat{\mathbf{X}} \in \mathbb{R}^{V \times N \times P}$, $\mathbf{W}_p \in \mathbb{R}^{P \times E}$, $\mathbf{W}_{pos} \in \mathbb{R}^{N \times E}$, and $\mathbf{Z}_0 \in \mathbb{R}^{V \times N \times E}$. $\mathbf{Z}_0$ is then fed into the CMamba encoder, consisting of $k$ CMamba blocks:

$$\mathbf{H}_l = \text{M-Mamba}(\mathbf{Z}_{l-1}),$$
$$\mathbf{Z}_l = \text{GDD-MLP}(\mathbf{H}_l) + \mathbf{Z}_{l-1}, \tag{11}$$

where $l = 1, ..., k$. The final prediction is generated through a linear projection of the output from the last CMamba block:

$$\hat{\mathbf{Y}} = \text{DeNorm}(\text{Flatten}(\text{Silu}(\mathbf{Z}_k))\mathbf{W}_{proj}), \tag{12}$$

where $\mathbf{W}_{proj} \in \mathbf{R}^{(N*E) \times T}$ and $\hat{\mathbf{Y}} \in \mathbb{R}^{V \times T}$. In the testing stage, the Channel Mixup module is excluded, and the model is evaluated directly on the original testing set.

## 5 EXPERIMENTS

**Datasets** We evaluate the performance of CMamba on seven well-established datasets: ETTm1, ETTm2, ETTh1, ETTh2, Electricity, Weather, and Traffic. All of these datasets are publicly available (Wu et al., 2021). We follow the public splits and apply zero-mean normalization to each dataset. More details about datasets are provided in Appendix A.1.

**Baselines** We select ten advanced models as our baselines, including (i) Linear-based models: DLinear (Zeng et al., 2023), RLinear (Li et al., 2023), TiDE (Das et al., 2023), TimeMixer (Wang et al., 2023); (ii) Transformer-based models: Crossformer (Zhang & Yan, 2022), PatchTST (Nie et al., 2022), iTransformer (Liu et al., 2023a); and (iii) Convolution-based models: MICN (Wang et al., 2022), TimesNet (Wu et al., 2022), ModernTCN (Luo & Wang, 2024).

**Implementation** We set the look-back window to $L = 96$ and report the Mean Squared Error (MSE) as well as the Mean Absolute Error (MAE) for four prediction lengths $T \in \{96, 192, 336, 720\}$. We reuse most of the baseline results from iTransformer (Liu et al., 2023a) but we rerun MICN (Wang et al., 2022), TimeMixer (Wang et al., 2023), and ModernTCN (Luo & Wang, 2024) due to their different experimental settings. All experiments are repeated three times, and we report the mean. More details about hyperparameters can be found in Appendix A.2.

Table 1: Average results of the long-term forecasting task with prediction lengths $T \in \{96, 192, 336, 720\}$. We fix the look-back window $L = 96$ and report the average performance of all prediction lengths. The best is highlighted in **red** and the runner-up in blue.

| Models | CMamba (Ours) | | ModernTCN (2024) | | iTransformer (2023a) | | TimeMixer (2023) | | RLinear (2023) | | PatchTST (2022) | | Crossformer (2022) | | TiDE (2023) | | TimesNet (2022) | | MICN (2022) | | DLinear (2023) | |
|---|---|---|---|---|---|---|---|---|---|---|---|---|---|---|---|---|---|---|---|---|---|---|
| Metric | MSE | MAE | MSE | MAE | MSE | MAE | MSE | MAE | MSE | MAE | MSE | MAE | MSE | MAE | MSE | MAE | MSE | MAE | MSE | MAE | MSE | MAE |
| ETTm1 | **0.376** | **0.379** | 0.386 | 0.401 | 0.407 | 0.410 | 0.384 | 0.397 | 0.414 | 0.407 | 0.387 | 0.400 | 0.513 | 0.496 | 0.419 | 0.419 | 0.400 | 0.406 | 0.407 | 0.432 | 0.403 | 0.407 |
| ETTm2 | **0.273** | **0.316** | 0.278 | 0.322 | 0.288 | 0.332 | 0.279 | 0.325 | 0.286 | 0.327 | 0.281 | 0.326 | 0.757 | 0.610 | 0.358 | 0.404 | 0.291 | 0.333 | 0.339 | 0.386 | 0.350 | 0.401 |
| ETTh1 | **0.433** | **0.425** | 0.445 | 0.432 | 0.454 | 0.447 | 0.470 | 0.451 | 0.446 | 0.434 | 0.469 | 0.454 | 0.529 | 0.522 | 0.541 | 0.507 | 0.458 | 0.450 | 0.559 | 0.524 | 0.456 | 0.452 |
| ETTh2 | **0.368** | **0.391** | 0.381 | 0.404 | 0.383 | 0.407 | 0.389 | 0.409 | 0.374 | 0.398 | 0.387 | 0.407 | 0.942 | 0.684 | 0.611 | 0.550 | 0.414 | 0.427 | 0.580 | 0.526 | 0.559 | 0.515 |
| Electricity | **0.169** | **0.258** | 0.197 | 0.282 | 0.178 | 0.270 | 0.183 | 0.272 | 0.219 | 0.298 | 0.216 | 0.304 | 0.244 | 0.334 | 0.251 | 0.344 | 0.192 | 0.295 | 0.185 | 0.296 | 0.212 | 0.300 |
| Weather | **0.237** | **0.259** | 0.240 | 0.271 | 0.258 | 0.279 | 0.245 | 0.274 | 0.272 | 0.291 | 0.259 | 0.281 | 0.259 | 0.315 | 0.271 | 0.320 | 0.259 | 0.287 | 0.267 | 0.318 | 0.265 | 0.317 |
| Traffic | 0.444 | **0.265** | 0.546 | 0.348 | **0.428** | 0.282 | 0.496 | 0.298 | 0.626 | 0.378 | 0.555 | 0.362 | 0.550 | 0.304 | 0.760 | 0.473 | 0.620 | 0.336 | 0.544 | 0.319 | 0.625 | 0.383 |

## 5.1 MAIN RESULTS

The comprehensive results for multivariate long-term time series forecasting are presented in Table 1. We report the average performance across four prediction horizons $T \in \{96, 192, 336, 720\}$ in the main text, with full results available in Appendix D.1. Compared to other state-of-the-art methods, CMamba ranks top 1 in **13** out of the 14 settings of varying metrics and top 2 in **all** settings. Actually, across all prediction lengths and metrics, encompassing 70 settings, CMamba ranks top 1 in **65** settings and top 2 in **all** settings (detailed in Appendix D.1). Notably, for datasets with numerous time series, such as Electricity, Weather, and Traffic, CMamba performs as well as or better than iTransformer. iTransformer captures cross-channel dependencies via the self-attention mechanism, incurring high computational costs. Our GDD-MLP module achieves comparable data dependencies and global receptive field, but the computational cost is significantly reduced. These results demonstrate our method's effectiveness. For experiments with the optimal look-back length, we provide the results in Appendix C.4.

## 5.2 ABLATION STUDY

**Ablation of M-Mamba Design** In Section 4.1.1, we describe the differences between the M-Mamba and vanilla Mamba modules, including the removal of convolution, feature-independent **A**, and data-dependent **D**. Here, we provide quantitative evidence to justify our design choices. As shown in Table 2, we conduct experiments on the Weather dataset with the same settings as before. Case Vanilla, ①, ②, and ③ indicate that the convolution operation and the gated z-branch are redundant for time series forecasting. However, removing both the convolution operation and the z-branch does

Table 2: Ablation of M-Mamba on Weather. **FS**: feature-specific, **FI**: feature-independent, **F**: free variables, **DD**: data-dependent variables.

| Case | Conv | z-branch | A | D | MSE | MAE |
|---|---|---|---|---|---|---|
| Vanilla | ✓ | ✓ | FS | F | 0.240 | 0.261 |
| ① | - | ✓ | FS | F | 0.238 | 0.260 |
| ② | ✓ | - | FS | F | 0.239 | 0.261 |
| ③ | - | - | FS | F | 0.239 | 0.261 |
| ④ | ✓ | ✓ | FI | F | 0.240 | 0.261 |
| ⑤ | ✓ | ✓ | FI | DD | 0.239 | 0.260 |
| CMamba | - | ✓ | FI | DD | **0.237** | **0.259** |

not provide further improvement, suggesting that retaining the z-branch is beneficial. Moreover, Case Vanilla and Case ④ demonstrate that learning a feature-specific transfer matrix **A** is unnecessary, given the similar temporal characteristics within each patch. On the other hand, due to the differences between channels and patches, making the skip connection matrix **D** data-dependent is justified, as indicated by the comparison between Case ④ and Case ⑤.

**Ablation of GDD-MLP and Channel Mixup** To assess the contributions of each module in CMamba, we conduct ablation studies on the GDD-MLP and Channel Mixup modules. The results, summarized in Table 3, present the average performance across four prediction horizons, with full results available in Appendix D.2. Overall, the combination of both modules leads to state-of-the-art performance, demonstrating the efficacy of their integration. In most cases, both modules could function independently, yielding significant improvements over baseline models. However, for the Traffic dataset, using GDD-MLP in isolation results in substantial performance degradation. This supports our hypothesis that the Channel-Dependent (CD) strategy, when applied without Channel Mixup, is vulnerable to distributional shifts and prone to overfitting.

Table 3: Ablation of GDD-MLP and Channel Mixup. We list the average MSE and MAE of different prediction lengths.

| GDD-MLP | Channel Mixup | ETTm1 | | ETTm2 | | ETTh1 | | ETTh2 | | Electricity | | Weather | | Traffic | |
|---|---|---|---|---|---|---|---|---|---|---|---|---|---|---|---|
| | | MSE | MAE | MSE | MAE | MSE | MAE | MSE | MAE | MSE | MAE | MSE | MAE | MSE | MAE |
| - | - | 0.387 | 0.387 | 0.282 | 0.322 | 0.431 | 0.428 | **0.367** | **0.391** | 0.190 | 0.268 | 0.255 | 0.271 | 0.479 | **0.264** |
| - | ✓ | 0.383 | 0.382 | 0.280 | 0.321 | **0.430** | 0.426 | 0.373 | 0.393 | 0.189 | 0.267 | 0.254 | 0.270 | 0.483 | 0.267 |
| ✓ | - | 0.388 | 0.387 | 0.275 | 0.317 | 0.437 | 0.428 | 0.372 | 0.394 | 0.175 | 0.266 | 0.241 | 0.262 | 0.525 | 0.285 |
| ✓ | ✓ | **0.376** | **0.379** | **0.273** | **0.316** | 0.433 | **0.425** | 0.368 | **0.391** | **0.169** | **0.258** | **0.237** | **0.259** | **0.444** | 0.265 |

## 5.3 MODEL ANALYSIS

**Effectiveness of GDD-MLP and Channel Mixup**    Here, we further demonstrate the advantages of our proposed GDD-MLP and Channel Mixup over traditional mixup and MLP by visualizing the optimization process, with the patch-wise M-Mamba as our base model. (1) As shown in Fig. 4 (a) and (b), compared to the vanilla mixup, Channel Mixup successfully suppresses the oversmoothing caused by the CD strategy and brings significantly excellent generalization capabilities. (2) As depicted in Fig. 4 (c) and (d), the traditional MLP does show a stronger fitting ability, but because it is position-dependent rather than data-dependent, its generalization ability is weaker than that of GDD-MLP. Moreover, due to the data-dependent mechanism, GDD-MLP is more compatible with Channel Mixup than traditional MLP.

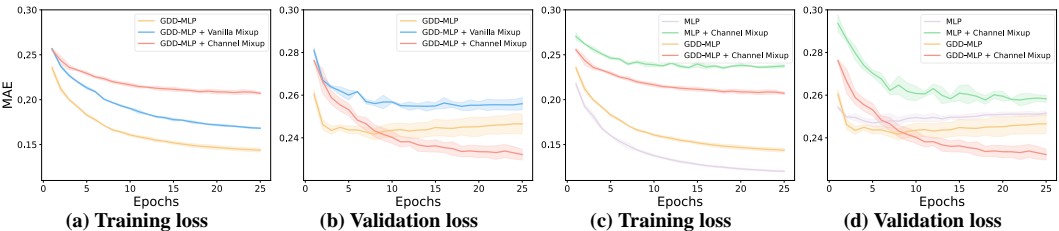

(a) Training loss    (b) Validation loss    (c) Training loss    (d) Validation loss

Figure 4: Loss curves for the Traffic dataset with look-back length and prediction length fixed at 96. All curves are generated by models that combine the specified module with M-Mamba. For example, GDD-MLP + Channel Mixup corresponds to CMamba.

**Generalizability of GDD-MLP and Channel Mixup**    We evaluate the effectiveness and versatility of GDD-MLP and Channel Mixup on four recent models: iTransformer (Liu et al., 2023a) and PatchTST (Nie et al., 2022) (Transformer-based), RLinear (Li et al., 2023) (Linear-based), and TimesNet (Wu et al., 2022) (Convolution-based). Among them, iTransformer and TimesNet adopt CD strategies, while PatchTST and RLinear utilize CI approaches. We retain the original architectures of these models but process the input via Channel Mixup during training and insert the GDD-MLP module into the original frameworks. The modified architectures are detailed in Appendix B.2. As shown in Table 4, our pipeline consistently enhances performance over various models, yielding an average improvement of **5**% across all metrics. For iTransformer and TimesNet, which have already taken cross-channel dependencies into account, the proposed modules do not result in major

Table 4: Performance promotion obtained by our proposed GDD-MLP and Channel Mixup when applying them to other frameworks. We fix the look-back window $L = 96$ and report the average performance of four prediction lengths $T \in \{96, 192, 336, 720\}$.

| Method | | iTransformer | | PatchTST | | RLinear | | TimesNet | |
|---|---|---|---|---|---|---|---|---|---|
| Metric | | MSE | MAE | MSE | MAE | MSE | MAE | MSE | MAE |
| Electricity | Original | 0.178 | 0.270 | 0.216 | 0.304 | 0.219 | 0.298 | 0.192 | 0.295 |
| | w/ GDD-MLP and Channel Mixup | **0.170** | **0.264** | **0.178** | **0.273** | **0.203** | **0.290** | **0.182** | **0.284** |
| | Promotion | **4.6%** | **2.1%** | **17.8%** | **10.2%** | **7.5%** | **2.6%** | **5.2%** | **3.7%** |
| Weather | Original | 0.258 | 0.279 | 0.259 | 0.281 | 0.272 | 0.291 | 0.259 | 0.287 |
| | w/ GDD-MLP and Channel Mixup | **0.250** | **0.274** | **0.248** | **0.272** | **0.258** | **0.285** | **0.257** | **0.282** |
| | Promotion | **3.1%** | **1.9%** | **4.4%** | **3.3%** | **5.1%** | **2.1%** | **0.9%** | **1.6%** |

improvements. However, for PatchTST and RLinear, which adopt CI strategies, the integration of our modules effectively mitigates oversmoothing, leading to substantial performance gains.

**Computational Cost of GDD-MLP**     We report the computational cost introduced by the GDD-MLP module in Table 5, quantified by FLOPs (G). To ensure a fair comparison, we conduct experiments on our CMamba with a fixed hidden size of 128, three layers, and a batch size of 64 for the ETT, Weather, and Electricity datasets. Due to memory limitations, the batch size of the Traffic dataset is set to 32. As shown in Table 5, the GDD-MLP module contributes minimally to the overall computational cost. Notably, even for the Traffic dataset, which includes 862 channels, the increase in FLOPs is only 1.35%, indicating the module's efficiency.

Table 5: Computational cost of GDD-MLP. We use FLOPs (G) to measure the computational complexity.

| Dataset
Channel | ETT
7 | Weather
21 | Electricity
321 | Traffic
862 |
|---|---|---|---|---|
| w/o GDD-MLP | 1.3322 | 3.9966 | 61.0916 | 82.0264 |
| w/ GDD-MLP | 1.3368 | 4.0130 | 61.7558 | 83.1327 |
| FLOPs increment | 0.35% | 0.41% | 1.09% | 1.35% |

**Longer Look-back Length**     The look-back length determines the extent of temporal information available within time series data. A model's ability to deliver improved forecasting performance with an extended historical window indicates its proficiency in capturing long-range temporal dependencies. Fig. 5 illustrates the performance trajectories of CMamba across varying look-back lengths. Consistent with other state-of-the-art models (Nie et al., 2022; Liu et al., 2023a), CMamba's performance improves as the look-back window lengthens, aligning with the hypothesis that an expanded receptive field contributes to more accurate predictions.

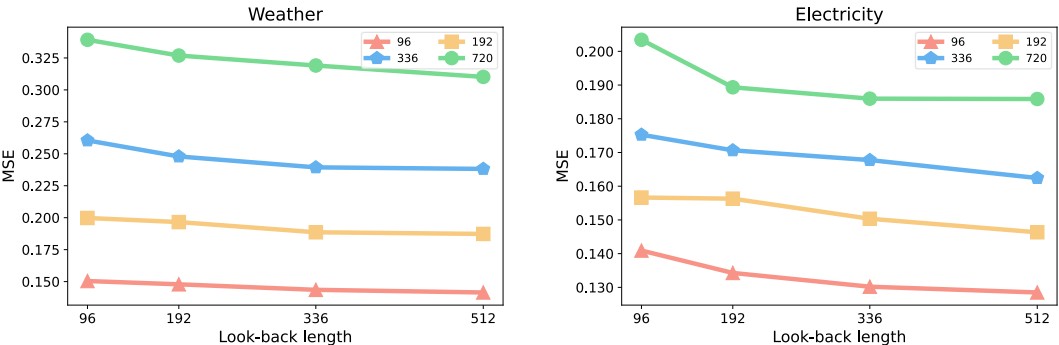

Figure 5: Performance promotion with longer look-back lengths. We vary the look-back length $L$ in $\{96, 192, 336, 512\}$ and report the performance curves of four prediction lengths $T \in \{96, 192, 336, 720\}$ under the Weather and Electricity dataset.

## 6  CONCLUSION

We propose CMamba, a novel state space model for multivariate time series forecasting. To balance cross-time and cross-channel dependencies, CMamba consists of three key components: a M-Mamba module that facilitates Mamba for cross-time dependencies modeling, a GDD-MLP module that captures cross-channel dependencies, and a Channel Mixup training strategy that enhances generalization and facilitates the CD approach. Extensive experiments demonstrate that CMamba achieves state-of-the-art performance on seven real-world datasets. Notably, the GDD-MLP and Channel Mixup modules could be seamlessly inserted into other models with minimal cost, showcasing remarkable framework versatility. In the future, we aim to explore more effective techniques to capture cross-time and cross-channel dependencies.

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

# A    IMPLEMENTATION DETAILS

## A.1    DATASET DESCRIPTIONS

We conduct experiments on seven real-world datasets following the setups in previous works (Liu et al., 2023a; Nie et al., 2022). (1) Four *ETT* (Electricity Transformer Temperature) datasets contain seven indicators from two different electric transformers in two years, each of which includes two different resolutions: 15 minutes (*ETTm1* and *ETTm2*) and 1 hour (*ETTh1* and *ETTh2*). (2) *Electricity* comprises the hourly electricity consumption of 321 customers in two years. (3) *Weather* contains 21 meteorological factors recorded every 10 minutes in Germany in 2020. (4) *Traffic* collects the hourly road occupancy rates from 862 different sensors on San Francisco freeways in two years. More details are provided in Table 6.

Table 6: Detailed dataset descriptions. $Channel$ indicates the number of variates. $Frequency$ denotes the sampling intervals of time steps. $Domain$ indicates the physical realm of each dataset. $Prediction\ Length$ denotes the future time points to be predicted. The last row indicates the ratio of training, validation, and testing sets.

| Dataset | Channel | Frequency | Domain | Prediction Length | Training:Validation:Testing |
|---------|---------|-----------|--------|-------------------|------------------------------|
| ETTm1 | 7 | 15 minutes | Electricity | {96, 192, 336, 720} | 6:2:2 |
| ETTm2 | 7 | 15 minutes | Electricity | {96, 192, 336, 720} | 6:2:2 |
| ETTh1 | 7 | 1 hour | Electricity | {96, 192, 336, 720} | 6:2:2 |
| ETTh2 | 7 | 1 hour | Electricity | {96, 192, 336, 720} | 6:2:2 |
| Electricity | 321 | 1 hour | Electricity | {96, 192, 336, 720} | 7:1:2 |
| Weather | 21 | 10 minutes | Weather | {96, 192, 336, 720} | 7:1:2 |
| Traffic | 862 | 1 hour | Transportation | {96, 192, 336, 720} | 7:1:2 |

## A.2    HYPERPARAMETERS

We conduct experiments on a single NVIDIA A100 40GB GPU. We utilize Adam (Kingma & Ba, 2014) optimizer with L1 loss and tune the initial learning rate in $\{0.0001, 0.0005, 0.001\}$. We fix the patch length at $16$ and the patch stride at $8$. The embedding of patches is selected from $\{128, 256\}$. The number of CMamba blocks is searched in $\{2, 3, 4, 5\}$. The standard deviation ($\sigma$) of Channel Mixup is tuned from $0.1$ to $5$. The dropout rate is searched in $\{0, 0.1\}$. For the M-Mamba module, we fix the dimension of the hidden state at $16$ and the expansion rate of the linear layer at $1$. To ensure robustness, we run our model three times under three random seeds (2020, 2021, 2022) in each setting. The average performance along with the standard deviation is presented in Table 7.

Table 7: Robustness of the proposed CMamba performance. The results are generated from three random seeds.

| Horizon | Dataset | ETTm1 | ETTm2 | ETTh1 | ETTh2 | Electricity | Weather | Traffic |
|---------|---------|-------|-------|-------|-------|-------------|---------|---------|
| 96 | MSE | 0.308±0.004 | 0.171±0.001 | 0.372±0.001 | 0.281±0.001 | 0.141±0.000 | 0.150±0.001 | 0.414±0.003 |
|    | MAE | 0.338±0.003 | 0.248±0.001 | 0.386±0.000 | 0.329±0.001 | 0.231±0.001 | 0.187±0.000 | 0.251±0.001 |
| 192 | MSE | 0.359±0.001 | 0.235±0.001 | 0.422±0.003 | 0.361±0.000 | 0.157±0.002 | 0.200±0.001 | 0.432±0.001 |
|     | MAE | 0.364±0.001 | 0.292±0.000 | 0.416±0.001 | 0.381±0.000 | 0.245±0.002 | 0.236±0.001 | 0.257±0.001 |
| 336 | MSE | 0.390±0.005 | 0.296±0.000 | 0.466±0.003 | 0.413±0.001 | 0.175±0.001 | 0.260±0.000 | 0.446±0.002 |
|     | MAE | 0.389±0.001 | 0.334±0.001 | 0.438±0.001 | 0.419±0.001 | 0.265±0.001 | 0.281±0.000 | 0.265±0.002 |
| 720 | MSE | 0.447±0.003 | 0.392±0.002 | 0.470±0.003 | 0.419±0.001 | 0.203±0.002 | 0.339±0.002 | 0.485±0.003 |
|     | MAE | 0.425±0.001 | 0.391±0.001 | 0.461±0.000 | 0.435±0.000 | 0.289±0.001 | 0.334±0.001 | 0.286±0.003 |

# B    BASELINES

## B.1    BASELINE DESCRIPTIONS

We carefully selected 10 state-of-the-art models for our study. Their details are as follows:

1) DLinear (Zeng et al., 2023) is a Linear-based model utilizing decomposition and a Channel-Independent strategy. The source code is available at `https://github.com/cure-lab/LTSF-Linear`.

2) MICN (Wang et al., 2022) is a Convolution-based model featuring multi-scale hybrid decomposition and multi-scale convolution. The source code is available at `https://github.com/wanghq21/MICN`.

3) TimesNet (Wu et al., 2022) decomposes 1D time series into 2D time series based on multi-periodicity and captures intra-period and inter-period correlations via convolution. The source code is available at `https://github.com/thuml/Time-Series-Library`.

4) TiDE (Das et al., 2023) adopts a pure MLP structure and a Channel-Independent strategy. The source code is available at `https://github.com/google-research/google-research/tree/master/tide`.

5) Crossformer (Zhang & Yan, 2022) is a patch-wise Transformer-based model with two-stage attention that captures cross-time and cross-channel dependencies, respectively. The source code is available at `https://github.com/Thinklab-SJTU/Crossformer`.

6) PatchTST (Nie et al., 2022) is a patch-wise Transformer-based model that adopts a Channel-Independent strategy. The source code is available at `https://github.com/yuqinie98/PatchTST`.

7) RLinear (Li et al., 2023) is a Linear-based model with RevIN and a Channel-Independent strategy. The source code is available at `https://github.com/plumprc/RTSF`.

8) TimeMixer (Wang et al., 2023) is a fully MLP-based model that leverages multiscale time series. It makes predictions based on the multiscale seasonal and trend information of time series. The source code is available at `https://github.com/kwuking/TimeMixer`.

9) iTransformer (Liu et al., 2023a) is an inverted Transformer-based model that captures cross-channel dependencies via the self-attention mechanism and cross-time dependencies via linear projection. The source code is available at `https://github.com/thuml/iTransformer`.

10) ModernTCN (Luo & Wang, 2024) is a Convolution-based model with larger receptive fields. It utilizes depth-wise convolution to learn the patch-wise temporal information and two point-wise convolution layers to capture cross-time and cross-channel dependencies respectively. The source code is available at `https://github.com/luodhhh/ModernTCN`.

Notably, the source code of most of these models is available at `https://github.com/thuml/Time-Series-Library` (Wang et al., 2024b).

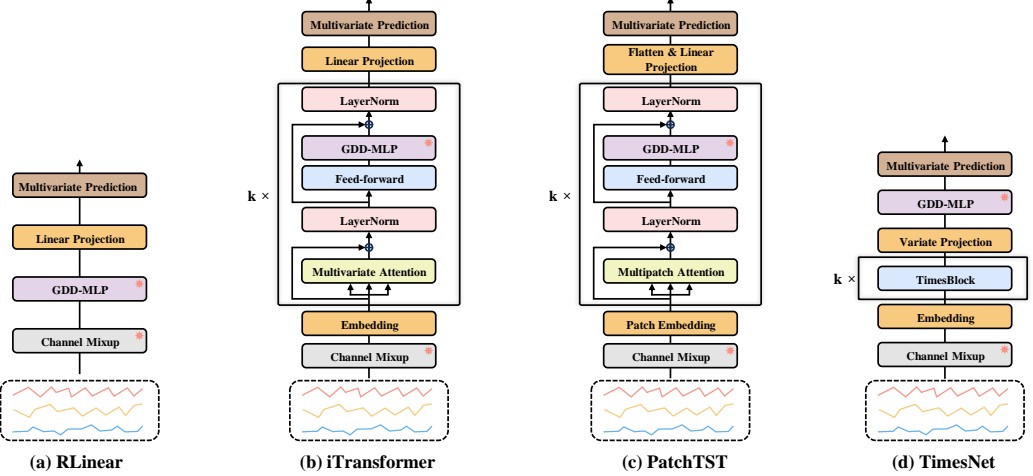

Figure 6: Modifications of the four chosen baselines. We retain the original architecture unchanged but apply Channel Mixup during training and insert the GDD-MLP module into the original model. The * modules represent GDD-MLP and Channel Mixup.

Table 8: Performance promotion obtained by our proposed GDD-MLP and Channel Mixup when applying them to other frameworks. We fix the look-back window $L = 96$ and report the performance of four prediction lengths $T \in \{96, 192, 336, 720\}$. Avg means the average metrics for four prediction lengths. ↑ indicates improved performance and ↓ denotes decreasing performance.

| Method | | | iTransformer | | PatchTST | | RLinear | | TimesNet | |
|---|---|---|---|---|---|---|---|---|---|---|
| Metric | | | MSE | MAE | MSE | MAE | MSE | MAE | MSE | MAE |
| Electricity | Original | 96 | 0.148 | 0.240 | 0.195 | 0.285 | 0.201 | 0.281 | 0.168 | 0.272 |
| | | 192 | 0.162 | 0.253 | 0.199 | 0.289 | 0.201 | 0.283 | 0.184 | 0.289 |
| | | 336 | 0.178 | 0.269 | 0.215 | 0.305 | 0.215 | 0.298 | 0.198 | 0.300 |
| | | 720 | 0.225 | 0.317 | 0.256 | 0.337 | 0.257 | 0.331 | 0.220 | 0.320 |
| | | Avg | 0.178 | 0.270 | 0.216 | 0.304 | 0.219 | 0.298 | 0.192 | 0.295 |
| | w/ GDD-MLP and Channel Mixup | 96 | 0.142↑ | 0.238↑ | 0.151↑ | 0.250↑ | 0.182↑ | 0.274↑ | 0.163↑ | 0.268↑ |
| | | 192 | 0.161↑ | 0.255↓ | 0.165↑ | 0.259↑ | 0.187↑ | 0.276↑ | 0.172↑ | 0.274↑ |
| | | 336 | 0.179↓ | 0.274↓ | 0.178↑ | 0.276↑ | 0.202↑ | 0.291↑ | 0.183↑ | 0.286↑ |
| | | 720 | 0.196↑ | 0.290↑ | 0.217↑ | 0.306↑ | 0.239↑ | 0.321↑ | 0.211↑ | 0.308↑ |
| | | Avg | 0.170↑ | 0.264↑ | 0.178↑ | 0.273↑ | 0.203↑ | 0.290↑ | 0.182↑ | 0.284↑ |
| | Promotion Count (Promotion / Total) | - | 4 / 5 | 3 / 5 | 5 / 5 | 5 / 5 | 5 / 5 | 5 / 5 | 5 / 5 | 5 / 5 |
| Weather | Original | 96 | 0.174 | 0.214 | 0.177 | 0.218 | 0.192 | 0.232 | 0.172 | 0.220 |
| | | 192 | 0.221 | 0.254 | 0.225 | 0.256 | 0.240 | 0.271 | 0.219 | 0.261 |
| | | 336 | 0.278 | 0.296 | 0.278 | 0.297 | 0.292 | 0.307 | 0.280 | 0.306 |
| | | 720 | 0.358 | 0.349 | 0.354 | 0.348 | 0.364 | 0.353 | 0.365 | 0.359 |
| | | Avg | 0.258 | 0.279 | 0.259 | 0.281 | 0.272 | 0.291 | 0.259 | 0.287 |
| | w/ GDD-MLP and Channel Mixup | 96 | 0.164↑ | 0.207↑ | 0.164↑ | 0.207↑ | 0.181↑ | 0.228↑ | 0.166↑ | 0.213↑ |
| | | 192 | 0.214↑ | 0.250↑ | 0.211↑ | 0.249↑ | 0.223↑ | 0.264↑ | 0.221↓ | 0.259↑ |
| | | 336 | 0.272↑ | 0.292↑ | 0.269↑ | 0.289↑ | 0.277↑ | 0.301↑ | 0.282↓ | 0.305↑ |
| | | 720 | 0.350↑ | 0.345↑ | 0.346↑ | 0.342↑ | 0.352↑ | 0.347↑ | 0.357↑ | 0.353↑ |
| | | Avg | 0.250↑ | 0.274↑ | 0.248↑ | 0.272↑ | 0.258↑ | 0.285↑ | 0.257↑ | 0.282↑ |
| | Promotion Count (Promotion / Total) | - | 5 / 5 | 5 / 5 | 5 / 5 | 5 / 5 | 5 / 5 | 5 / 5 | 3 / 5 | 5 / 5 |

## B.2 BASELINE MODIFICATION

In Section 5.3, we evaluate the effects of GDD-MLP and Channel Mixup modules on four state-of-the-art models: iTransformer (Liu et al., 2023a) and PatchTST (Nie et al., 2022) (Transformer-based), RLinear (Li et al., 2023) (Linear-based), and TimesNet (Wu et al., 2022) (Convolution-based). During experiments, we retain the original architecture unchanged but process the input via Channel Mixup during training and insert the GDD-MLP module into the original model. The modified frameworks of these models are shown in Fig. 6. All models adopt instance norm or RevIN (Kim et al., 2021) based on their original settings. We only tune the standard deviation $\sigma$, and learning rate $lr$ for optimal performance. The full results are shown in Table 8.

## C MORE EVALUATION

### C.1 FULL ABLATION OF MAMBA DESIGN

In the main text, we assess the effectiveness of M-Mamba on the Weather dataset. Here, we present a more comprehensive evaluation across all datasets and prediction lengths. As detailed in Table 9, M-Mamba achieves performance gains in **58** out of 70 scenarios, with improvements becoming increasingly prominent at longer prediction horizons. This phenomenon is consistent with our architectural adjustments: removing the convolution operation and adopting a feature-independent matrix **A**, which deemphasizes local temporal dependencies, allowing Mamba to better capture long-term dependencies. Recent studies, including iTransformer (Liu et al., 2023a) and ModernTCN (Luo & Wang, 2024), underscore the importance of a global receptive field and long-term dependency modeling for effective long-term time series forecasting. Specifically, iTransformer utilizes self-attention to achieve this, while ModernTCN leverages large convolution kernels. By removing modules emphasizing local dependencies, we enable Mamba to prioritize global dependency capture, as evidenced by the significant improvements in experimental results. These findings validate the effectiveness of our modifications in enhancing Mamba's capacity for long-term forecasting.

Table 9: Comparisons of the vanilla Mamba and M-Mamba under all datasets and prediction lengths. Avg means the average metrics for four prediction lengths. ↑ indicates improved performance and ↓ denotes decreasing performance.

| | Dataset | ETTm1 | | ETTm2 | | ETTh1 | | ETTh2 | | Electricity | | Weather | | Traffic | |
|---|---|---|---|---|---|---|---|---|---|---|---|---|---|---|---|
| | Metric | MSE | MAE | MSE | MAE | MSE | MAE | MSE | MAE | MSE | MAE | MSE | MAE | MSE | MAE |
| Mamba | 96 | 0.312 | 0.341 | 0.173 | 0.251 | 0.376 | 0.390 | 0.283 | 0.331 | 0.140 | 0.231 | 0.152 | 0.189 | 0.404 | 0.250 |
| | 192 | 0.365 | 0.367 | 0.239 | 0.296 | 0.423 | 0.417 | 0.360 | 0.381 | 0.159 | 0.247 | 0.201 | 0.236 | 0.429 | 0.255 |
| | 336 | 0.398 | 0.392 | 0.300 | 0.336 | 0.468 | 0.439 | 0.416 | 0.422 | 0.177 | 0.265 | 0.264 | 0.283 | 0.455 | 0.265 |
| | 720 | 0.455 | 0.427 | 0.398 | 0.393 | 0.474 | 0.466 | 0.419 | 0.437 | 0.214 | 0.298 | 0.344 | 0.337 | 0.483 | 0.286 |
| | Avg | 0.383 | 0.381 | 0.277 | 0.319 | 0.435 | 0.428 | 0.370 | 0.393 | 0.173 | 0.260 | 0.240 | 0.261 | 0.443 | 0.264 |
| M-Mamba | 96 | 0.308↑ | 0.338↑ | 0.171↑ | 0.248↑ | 0.372↑ | 0.386↑ | 0.281↑ | 0.329↑ | 0.141↑ | 0.231↓ | 0.150↑ | 0.187↑ | 0.414↓ | 0.251↓ |
| | 192 | 0.359↑ | 0.364↑ | 0.235↑ | 0.292↑ | 0.422↑ | 0.416↑ | 0.361↓ | 0.381↓ | 0.157↑ | 0.245↑ | 0.200↑ | 0.236↑ | 0.432↓ | 0.257↓ |
| | 336 | 0.390↑ | 0.389↑ | 0.296↑ | 0.334↑ | 0.466↑ | 0.438↑ | 0.413↑ | 0.419↑ | 0.175↑ | 0.265↑ | 0.260↑ | 0.281↑ | 0.446↑ | 0.265↑ |
| | 720 | 0.447↑ | 0.425↑ | 0.392↑ | 0.391↑ | 0.470↑ | 0.461↑ | 0.419↑ | 0.435↑ | 0.203↑ | 0.289↑ | 0.339↑ | 0.334↑ | 0.485↓ | 0.286↑ |
| | Avg | 0.376↑ | 0.379↑ | 0.273↑ | 0.316↑ | 0.433↑ | 0.425↑ | 0.368↑ | 0.391↑ | 0.169↑ | 0.258↑ | 0.237↑ | 0.259↑ | 0.444↓ | 0.265↓ |
| Promotion Count | | 5 / 5 | 5 / 5 | 5 / 5 | 5 / 5 | 5 / 5 | 5 / 5 | 4 / 5 | 4 / 5 | 4 / 5 | 4 / 5 | 5 / 5 | 5 / 5 | 1 / 5 | 1 / 5 |

## C.2 EFFICIENCY ANALYSIS

In this section, we evaluate the efficiency of CMamba on the Weather and Electricity datasets, each with 21 and 321 channels, respectively. Both training time and model size are taken into account, as shown in Fig. 7. We can find that CMamba is the most lightweight model on average except for the Linear-based model (e.g., RLinear and TimeMixer). This indicates that CMamba's excellent performance stems more from its capacity to capture historical time series patterns than from an increase in parameters. However, CMamba's running speed is not outstanding, which is primarily due to the suboptimal parallelism inherent to state space models (SSMs) compared to attention-based architectures. Nevertheless, CMamba's strong performance and lightweight model size still make it a highly efficient model, effectively balancing accuracy and computational efficiency.

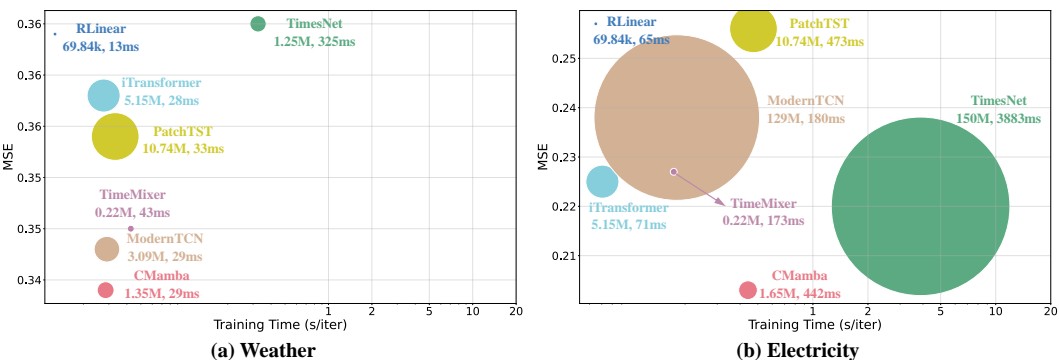

Figure 7: Efficiency comparison on the (a) Weather and (b) Electricity dataset. The look-back window is 96 and the horizon is 720.

## C.3 HYPERPARAMETER SENSITIVITY

We conduct a thorough evaluation of the hyperparameters influencing CMamba's performance, focusing on two critical factors: the standard deviation ($\sigma$) in the Channel Mixup mechanism and the expansion rate ($r$) for the GDD-MLP. Experiments are performed on the ETTh1 and ETTm1 datasets, maintaining a fixed look-back window and forecasting horizon of 96. All other hyperparameters remain consistent with those reported in Table 1. The results, visualized in Fig. 8, highlight the following: (1) The standard deviation ($\sigma$) controls the extent of perturbation and mixing across channels, directly influencing the robustness of the model. Proper tuning is essential, as it governs the balance between stability and generalization. A standard normal distribution tends to yield stable performance across various scenarios, likely due to its balance in perturbation magnitude. (2) The expansion rate ($r$) controls the dimensionality of the hidden layers in the GDD-MLP. While larger hidden layers increase computational demands, they may also detrimentally affect model performance. This suggests that excessive emphasis on cross-channel dependencies can diminish the

model's ability to capture cross-temporal dynamics, which are more critical for multivariate time series forecasting. Conversely, an expansion rate that is too small can result in underfitting, as the model may lack sufficient capacity to capture the necessary dependencies across channels.

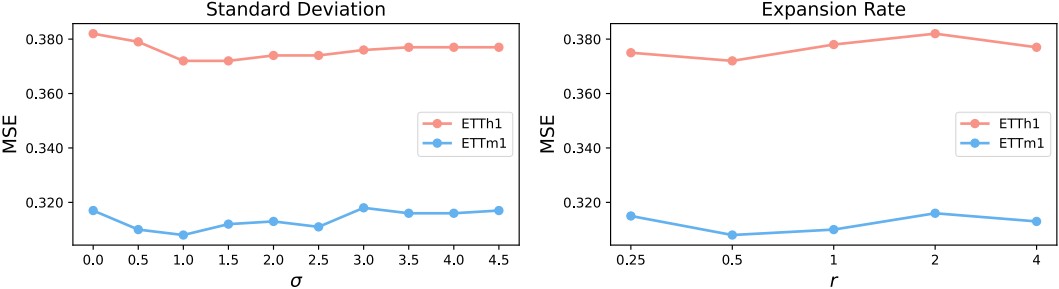

Figure 8: Hyperparameter sensitivity of the standard deviation $\sigma$ for Channel Mixup (left) and the expansion rate $r$ for GDD-MLP (right).

### C.4 UPPER BOUND EVALUATION

Considering that the performance of different models is influenced by the look-back length, we further compare our model with state-of-the-art frameworks under the **optimal** look-back length. As shown in Table 10, we compare the performance of each model using their best look-back window. For CMamba, we search the look-back length in $\{96, 192, 336, 512\}$. For other benchmarks, we rerun iTransformer since its look-back length is fixed at 96 in the original paper, and we collect results for other models from tables in ModernTCN (Luo & Wang, 2024), TimeMixer (Wang et al., 2023), and TiDE (Das et al., 2023). The results indicate that our model still achieves state-of-the-art performance.

Table 10: Full results of the long-term forecasting task under the **optimal** look-back window. We search the look-back window of CMamba in $\{96, 192, 336, 512\}$. We report the performance of four prediction lengths $T \in \{96, 192, 336, 720\}$. Avg means the average metrics for four prediction lengths. The best is highlighted in **red** and the runner-up in blue.

| Models | | CMamba (Ours) | | ModernTCN (2024) | | iTransformer (2023a) | | TimeMixer (2023) | | RLinear (2023) | | PatchTST (2022) | | Crossformer (2022) | | TiDE (2023) | | TimesNet (2022) | | MICN (2022) | | DLinear (2023) | |
|---|---|---|---|---|---|---|---|---|---|---|---|---|---|---|---|---|---|---|---|---|---|---|---|
| Metric | | MSE | MAE | MSE | MAE | MSE | MAE | MSE | MAE | MSE | MAE | MSE | MAE | MSE | MAE | MSE | MAE | MSE | MAE | MSE | MAE | MSE | MAE |
| ETTm1 | 96 | **0.283** | **0.329** | 0.292 | 0.346 | 0.295 | 0.344 | 0.291 | 0.340 | 0.301 | 0.342 | 0.290 | 0.342 | 0.316 | 0.373 | 0.306 | 0.349 | 0.338 | 0.375 | 0.314 | 0.360 | 0.299 | 0.343 |
| | 192 | 0.332 | **0.357** | 0.332 | 0.368 | 0.336 | 0.370 | **0.327** | 0.365 | 0.335 | 0.363 | 0.332 | 0.369 | 0.377 | 0.411 | 0.335 | 0.366 | 0.371 | 0.387 | 0.359 | 0.387 | 0.335 | 0.365 |
| | 336 | **0.357** | **0.378** | 0.365 | 0.391 | 0.373 | 0.391 | 0.360 | 0.381 | 0.370 | 0.383 | 0.366 | 0.392 | 0.431 | 0.442 | 0.364 | 0.384 | 0.410 | 0.411 | 0.398 | 0.413 | 0.369 | 0.386 |
| | 720 | 0.414 | **0.410** | 0.416 | 0.417 | 0.432 | 0.427 | 0.415 | 0.417 | 0.425 | 0.414 | 0.416 | 0.420 | 0.600 | 0.547 | 0.413 | 0.413 | 0.478 | 0.450 | 0.459 | 0.464 | 0.425 | 0.421 |
| | Avg | **0.347** | **0.368** | 0.351 | 0.381 | 0.359 | 0.383 | 0.348 | 0.375 | 0.358 | 0.376 | 0.351 | 0.381 | 0.431 | 0.443 | 0.355 | 0.378 | 0.400 | 0.406 | 0.383 | 0.406 | 0.357 | 0.379 |
| ETTh1 | 96 | **0.358** | **0.387** | 0.368 | 0.394 | 0.386 | 0.405 | 0.361 | 0.390 | 0.366 | 0.391 | 0.370 | 0.399 | 0.386 | 0.429 | 0.375 | 0.398 | 0.384 | 0.402 | 0.396 | 0.427 | 0.375 | 0.399 |
| | 192 | **0.399** | 0.415 | 0.405 | 0.413 | 0.441 | 0.436 | 0.409 | 0.414 | 0.404 | **0.412** | 0.413 | 0.421 | 0.419 | 0.444 | 0.412 | 0.422 | 0.436 | 0.429 | 0.430 | 0.453 | 0.405 | 0.416 |
| | 336 | 0.427 | 0.437 | **0.391** | **0.412** | 0.461 | 0.452 | 0.430 | 0.429 | 0.420 | 0.423 | 0.422 | 0.436 | 0.440 | 0.461 | 0.435 | 0.433 | 0.491 | 0.469 | 0.433 | 0.458 | 0.439 | 0.443 |
| | 720 | **0.428** | **0.451** | 0.450 | 0.461 | 0.503 | 0.491 | 0.445 | 0.460 | 0.442 | 0.456 | 0.447 | 0.466 | 0.519 | 0.524 | 0.454 | 0.465 | 0.521 | 0.500 | 0.474 | 0.508 | 0.472 | 0.490 |
| | Avg | **0.403** | 0.422 | 0.404 | 0.420 | 0.448 | 0.446 | 0.411 | 0.423 | 0.408 | 0.421 | 0.413 | 0.431 | 0.441 | 0.465 | 0.419 | 0.430 | 0.458 | 0.450 | 0.433 | 0.462 | 0.423 | 0.437 |
| Electricity | 96 | **0.128** | **0.220** | 0.129 | 0.226 | 0.132 | 0.228 | 0.129 | 0.224 | 0.140 | 0.235 | 0.129 | 0.222 | 0.219 | 0.314 | 0.132 | 0.229 | 0.168 | 0.272 | 0.159 | 0.267 | 0.153 | 0.237 |
| | 192 | 0.146 | 0.239 | 0.143 | 0.239 | 0.154 | 0.247 | **0.140** | **0.220** | 0.154 | 0.248 | 0.147 | 0.240 | 0.231 | 0.322 | 0.147 | 0.243 | 0.184 | 0.289 | 0.168 | 0.279 | 0.152 | 0.249 |
| | 336 | 0.162 | 0.258 | **0.161** | 0.259 | 0.172 | 0.266 | **0.161** | 0.255 | 0.171 | 0.264 | 0.163 | 0.259 | 0.246 | 0.337 | **0.161** | 0.261 | 0.198 | 0.300 | 0.196 | 0.308 | 0.169 | 0.267 |
| | 720 | **0.186** | **0.278** | 0.191 | 0.286 | 0.210 | 0.303 | 0.194 | 0.287 | 0.209 | 0.297 | 0.197 | 0.290 | 0.280 | 0.363 | 0.196 | 0.294 | 0.220 | 0.320 | 0.203 | 0.312 | 0.233 | 0.344 |
| | Avg | **0.156** | **0.249** | 0.156 | 0.253 | 0.167 | 0.261 | 0.156 | 0.246 | 0.169 | 0.261 | 0.159 | 0.253 | 0.244 | 0.334 | 0.159 | 0.257 | 0.192 | 0.295 | 0.182 | 0.292 | 0.177 | 0.274 |
| Weather | 96 | **0.141** | **0.180** | 0.149 | 0.200 | 0.162 | 0.212 | 0.147 | 0.197 | 0.175 | 0.225 | 0.149 | 0.198 | 0.153 | 0.217 | 0.166 | 0.222 | 0.172 | 0.220 | 0.161 | 0.226 | 0.152 | 0.237 |
| | 192 | **0.187** | **0.229** | 0.196 | 0.245 | 0.204 | 0.252 | 0.189 | 0.239 | 0.218 | 0.260 | 0.194 | 0.241 | 0.197 | 0.269 | 0.209 | 0.263 | 0.219 | 0.261 | 0.220 | 0.283 | 0.220 | 0.282 |
| | 336 | 0.238 | **0.267** | 0.238 | 0.277 | 0.256 | 0.290 | 0.241 | 0.280 | 0.265 | 0.294 | 0.245 | 0.282 | 0.252 | 0.311 | 0.254 | 0.301 | 0.280 | 0.306 | 0.275 | 0.328 | 0.265 | 0.319 |
| | 720 | **0.310** | **0.321** | 0.314 | 0.334 | 0.326 | 0.338 | 0.310 | 0.330 | 0.329 | 0.339 | 0.314 | 0.334 | 0.318 | 0.363 | 0.313 | 0.340 | 0.365 | 0.359 | 0.311 | 0.356 | 0.323 | 0.362 |
| | Avg | **0.219** | **0.249** | 0.224 | 0.264 | 0.237 | 0.273 | 0.222 | 0.262 | 0.247 | 0.279 | 0.226 | 0.264 | 0.230 | 0.290 | 0.236 | 0.282 | 0.259 | 0.287 | 0.242 | 0.298 | 0.240 | 0.300 |

### C.5 EFFECTIVENESS OF MAE LOSS

The choice of the loss function plays a pivotal role in both the optimization process and the convergence behavior of time series forecasting models. While many prior models predominantly employ L2 loss, also known as Mean Squared Error (MSE), our proposed model adopts L1 loss, also referred to as Mean Absolute Error (MAE). The rationale behind this choice stems from empirical observations across various datasets, where the mean absolute error is typically small. In such cases, the gradients produced by L2 loss tend to be small, which can impede the model's ability to escape

local minima during training, as depicted in Fig. 9. Moreover, time series data often exhibit outliers, and L2 loss is known to be more sensitive to these outliers, which can lead to instability during training. In contrast, L1 loss is more robust to outliers, promoting greater stability in model training and convergence.

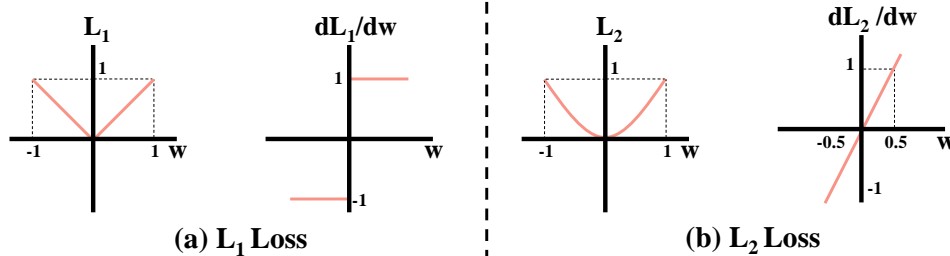

Figure 9: Visualization of L1 and L2 loss.

To substantiate this choice, we evaluate the performance of five state-of-the-art models from different architectural families: CMamba (our proposed model, SSM-based), ModernTCN (Convolution-based), iTransformer (Transformer-based), TimeMixer (Linear-based), and PatchTST (Transformer-based). We assess the performance of these models using both MSE and MAE loss functions. The results are exhibited in Table 11, where **37** out of 40 cases have improved performance after switching from MSE loss to MAE loss. Even when considering different loss functions, our model consistently outperforms the alternatives, achieving the best average performance, which underscores the effectiveness and robustness of CMamba under various training conditions.

Table 11: Performance promotion obtained by changing MSE loss to MAE loss. We fix the look-back window $L = 96$ and report the average performance of four prediction lengths $T \in \{96, 192, 336, 720\}$. The best under each loss is highlighted in **red**. ↑ indicates improved performance and ↓ denotes decreasing performance.

| Method | | CMamba | | ModernTCN | | iTransformer | | TimeMixer | | PatchTST | |
|---|---|---|---|---|---|---|---|---|---|---|---|
| Metric | | MSE | MAE | MSE | MAE | MSE | MAE | MSE | MAE | MSE | MAE |
| ETTm2 | MSE Loss | **0.278** | 0.324 | **0.278** | **0.322** | 0.288 | 0.332 | 0.279 | 0.325 | 0.281 | 0.326 |
| | MAE Loss | **0.273**↑ | **0.316**↑ | 0.276↑ | 0.317↑ | 0.283↑ | 0.322↑ | 0.274↑ | 0.317↑ | 0.279↑ | 0.318↑ |
| ETTh2 | MSE Loss | **0.380** | **0.403** | 0.381 | 0.404 | 0.383 | 0.407 | 0.389 | 0.409 | 0.387 | 0.407 |
| | MAE Loss | 0.368↑ | **0.391**↑ | 0.379↑ | 0.398↑ | 0.379↑ | 0.400↑ | 0.377↑ | 0.396↑ | **0.363**↑ | **0.391**↑ |
| Electricity | MSE Loss | **0.172** | **0.265** | 0.197 | 0.282 | 0.178 | 0.270 | 0.183 | 0.272 | 0.216 | 0.304 |
| | MAE Loss | **0.169**↑ | **0.258**↑ | 0.211↓ | 0.290↓ | 0.175↑ | 0.259↑ | 0.185↓ | 0.271↑ | 0.206↑ | 0.285↑ |
| Weather | MSE Loss | **0.240** | **0.270** | **0.240** | 0.271 | 0.258 | 0.279 | 0.245 | 0.274 | 0.259 | 0.281 |
| | MAE Loss | 0.237↑ | **0.259**↑ | **0.236**↑ | 0.263↑ | 0.255↑ | 0.271↑ | 0.245↑ | 0.265↑ | 0.255↑ | 0.270↑ |

## C.6 Limitations

In this work, we mainly focus on the multivariate time series forecasting task with endogenous variables, meaning that the values we aim to predict and the values treated as features only differ in terms of time steps. However, real-world scenarios often involve the influence of exogenous variables on the variables we seek to predict, a topic extensively discussed in prior research (Wang et al., 2024c). In addition, the experimental results show that our model exhibits significant improvements on some datasets with large-scale channels, such as Weather and Electricity. However, the improvements in MAE are relatively limited on the Traffic dataset, which contains 862 channels. This discrepancy could be attributed to the pronounced periodicity observed in traffic data compared to other domains. These periodic patterns are highly time-dependent, causing different channels to exhibit similar characteristics and obscuring their physical interconnections. Therefore, incorporating external variables and utilizing prior knowledge about the relationships between channels, such as the connectivity of traffic roads, might further enhance the prediction accuracy.

Table 12: Full results of the long-term forecasting task. We fix the look-back window $L = 96$ and make predictions for $T = \{96, 192, 336, 720\}$. Avg means the average metrics for four prediction lengths. The best is highlighted in **red** and the runner-up in blue. We report the average results of three random seeds.

| Models | | CMamba (Ours) | | ModernTCN (2024) | | iTransformer (2023a) | | TimeMixer (2023) | | RLinear (2023) | | PatchTST (2022) | | Crossformer (2022) | | TiDE (2023) | | TimesNet (2022) | | MICN (2022) | | DLinear (2023) | |
|---|---|---|---|---|---|---|---|---|---|---|---|---|---|---|---|---|---|---|---|---|---|---|---|
| Metric | | MSE | MAE | MSE | MAE | MSE | MAE | MSE | MAE | MSE | MAE | MSE | MAE | MSE | MAE | MSE | MAE | MSE | MAE | MSE | MAE | MSE | MAE |
| ETTm1 | 96 | **0.308** | **0.338** | 0.317 | 0.362 | 0.334 | 0.368 | 0.320 | 0.355 | 0.355 | 0.376 | 0.329 | 0.367 | 0.404 | 0.426 | 0.364 | 0.387 | 0.338 | 0.375 | 0.317 | 0.367 | 0.345 | 0.372 |
| | 192 | **0.359** | **0.364** | 0.363 | 0.389 | 0.377 | 0.391 | 0.362 | 0.382 | 0.391 | 0.392 | 0.367 | 0.385 | 0.450 | 0.451 | 0.398 | 0.404 | 0.374 | 0.387 | 0.382 | 0.413 | 0.380 | 0.389 |
| | 336 | **0.390** | **0.389** | 0.403 | 0.412 | 0.426 | 0.420 | 0.396 | 0.406 | 0.424 | 0.415 | 0.399 | 0.410 | 0.532 | 0.515 | 0.428 | 0.425 | 0.410 | 0.411 | 0.417 | 0.443 | 0.413 | 0.413 |
| | 720 | **0.447** | **0.425** | 0.461 | 0.443 | 0.491 | 0.459 | 0.458 | 0.445 | 0.487 | 0.450 | 0.454 | 0.439 | 0.666 | 0.589 | 0.487 | 0.461 | 0.478 | 0.450 | 0.511 | 0.505 | 0.474 | 0.453 |
| | Avg | **0.376** | **0.379** | 0.386 | 0.401 | 0.407 | 0.410 | 0.384 | 0.397 | 0.414 | 0.407 | 0.387 | 0.400 | 0.513 | 0.496 | 0.419 | 0.419 | 0.400 | 0.406 | 0.407 | 0.432 | 0.403 | 0.407 |
| ETTm2 | 96 | **0.171** | **0.248** | 0.173 | 0.255 | 0.180 | 0.264 | 0.176 | 0.259 | 0.182 | 0.265 | 0.175 | 0.259 | 0.287 | 0.366 | 0.207 | 0.305 | 0.187 | 0.267 | 0.182 | 0.278 | 0.193 | 0.292 |
| | 192 | **0.235** | **0.292** | 0.235 | 0.296 | 0.250 | 0.309 | 0.242 | 0.303 | 0.246 | 0.304 | 0.241 | 0.302 | 0.414 | 0.492 | 0.290 | 0.364 | 0.249 | 0.309 | 0.288 | 0.357 | 0.284 | 0.362 |
| | 336 | **0.296** | **0.334** | 0.308 | 0.344 | 0.311 | 0.348 | 0.303 | 0.339 | 0.307 | 0.342 | 0.305 | 0.343 | 0.597 | 0.542 | 0.377 | 0.422 | 0.321 | 0.351 | 0.370 | 0.413 | 0.369 | 0.427 |
| | 720 | **0.392** | **0.391** | 0.398 | 0.394 | 0.412 | 0.407 | 0.396 | 0.399 | 0.407 | 0.398 | 0.402 | 0.400 | 1.730 | 1.042 | 0.558 | 0.524 | 0.408 | 0.403 | 0.519 | 0.495 | 0.554 | 0.522 |
| | Avg | **0.273** | **0.316** | 0.278 | 0.322 | 0.288 | 0.332 | 0.279 | 0.325 | 0.286 | 0.327 | 0.281 | 0.326 | 0.757 | 0.610 | 0.358 | 0.404 | 0.291 | 0.333 | 0.339 | 0.386 | 0.350 | 0.401 |
| ETTh1 | 96 | **0.372** | **0.386** | 0.386 | 0.394 | 0.386 | 0.405 | 0.384 | 0.400 | 0.386 | 0.395 | 0.414 | 0.419 | 0.423 | 0.448 | 0.479 | 0.464 | 0.384 | 0.402 | 0.417 | 0.436 | 0.386 | 0.400 |
| | 192 | **0.422** | **0.416** | 0.436 | 0.423 | 0.441 | 0.436 | 0.437 | 0.429 | 0.439 | 0.424 | 0.460 | 0.445 | 0.471 | 0.474 | 0.525 | 0.492 | 0.436 | 0.429 | 0.488 | 0.476 | 0.437 | 0.432 |
| | 336 | **0.466** | **0.438** | 0.479 | 0.445 | 0.487 | 0.458 | 0.472 | 0.446 | 0.479 | 0.446 | 0.501 | 0.466 | 0.570 | 0.546 | 0.565 | 0.515 | 0.491 | 0.469 | 0.599 | 0.549 | 0.481 | 0.459 |
| | 720 | **0.470** | **0.461** | 0.481 | 0.469 | 0.503 | 0.491 | 0.586 | 0.531 | 0.481 | 0.470 | 0.500 | 0.488 | 0.653 | 0.621 | 0.594 | 0.558 | 0.521 | 0.500 | 0.730 | 0.634 | 0.519 | 0.516 |
| | Avg | **0.433** | **0.425** | 0.445 | 0.432 | 0.454 | 0.447 | 0.470 | 0.451 | 0.446 | 0.434 | 0.469 | 0.454 | 0.529 | 0.522 | 0.541 | 0.507 | 0.458 | 0.450 | 0.559 | 0.524 | 0.456 | 0.452 |
| ETTh2 | 96 | **0.281** | **0.329** | 0.292 | 0.340 | 0.297 | 0.349 | 0.297 | 0.348 | 0.288 | 0.338 | 0.302 | 0.348 | 0.745 | 0.584 | 0.400 | 0.440 | 0.340 | 0.374 | 0.355 | 0.402 | 0.333 | 0.387 |
| | 192 | **0.361** | **0.381** | 0.377 | 0.395 | 0.380 | 0.400 | 0.369 | 0.392 | 0.374 | 0.390 | 0.388 | 0.400 | 0.877 | 0.656 | 0.528 | 0.509 | 0.402 | 0.414 | 0.511 | 0.491 | 0.477 | 0.476 |
| | 336 | **0.413** | **0.419** | 0.424 | 0.434 | 0.428 | 0.432 | 0.427 | 0.435 | 0.415 | 0.426 | 0.426 | 0.433 | 1.043 | 0.731 | 0.643 | 0.571 | 0.452 | 0.452 | 0.618 | 0.551 | 0.594 | 0.541 |
| | 720 | **0.419** | **0.435** | 0.433 | 0.448 | 0.427 | 0.445 | 0.462 | 0.463 | 0.420 | 0.440 | 0.431 | 0.446 | 1.104 | 0.763 | 0.874 | 0.679 | 0.462 | 0.468 | 0.835 | 0.660 | 0.831 | 0.657 |
| | Avg | **0.368** | **0.391** | 0.381 | 0.404 | 0.383 | 0.407 | 0.389 | 0.409 | 0.374 | 0.398 | 0.387 | 0.407 | 0.942 | 0.684 | 0.611 | 0.550 | 0.414 | 0.427 | 0.580 | 0.526 | 0.559 | 0.515 |
| Electricity | 96 | **0.141** | **0.231** | 0.173 | 0.260 | 0.148 | 0.240 | 0.153 | 0.244 | 0.201 | 0.281 | 0.195 | 0.285 | 0.219 | 0.314 | 0.237 | 0.329 | 0.168 | 0.272 | 0.172 | 0.285 | 0.197 | 0.282 |
| | 192 | **0.157** | **0.245** | 0.181 | 0.267 | 0.162 | 0.253 | 0.168 | 0.259 | 0.201 | 0.283 | 0.199 | 0.289 | 0.231 | 0.322 | 0.236 | 0.330 | 0.184 | 0.289 | 0.177 | 0.287 | 0.196 | 0.285 |
| | 336 | **0.175** | **0.265** | 0.196 | 0.283 | 0.178 | 0.269 | 0.185 | 0.275 | 0.215 | 0.298 | 0.215 | 0.305 | 0.246 | 0.337 | 0.249 | 0.344 | 0.198 | 0.300 | 0.186 | 0.297 | 0.209 | 0.301 |
| | 720 | **0.203** | **0.289** | 0.238 | 0.316 | 0.225 | 0.319 | 0.227 | 0.312 | 0.257 | 0.331 | 0.256 | 0.337 | 0.280 | 0.363 | 0.284 | 0.373 | 0.220 | 0.320 | 0.204 | 0.314 | 0.245 | 0.333 |
| | Avg | **0.169** | **0.258** | 0.197 | 0.282 | 0.178 | 0.270 | 0.183 | 0.272 | 0.219 | 0.298 | 0.216 | 0.304 | 0.244 | 0.334 | 0.251 | 0.344 | 0.192 | 0.295 | 0.185 | 0.296 | 0.212 | 0.300 |
| Weather | 96 | **0.150** | **0.187** | 0.155 | 0.203 | 0.174 | 0.214 | 0.162 | 0.208 | 0.192 | 0.232 | 0.177 | 0.218 | 0.158 | 0.230 | 0.202 | 0.261 | 0.172 | 0.220 | 0.194 | 0.253 | 0.196 | 0.255 |
| | 192 | **0.200** | **0.236** | 0.202 | 0.247 | 0.221 | 0.254 | 0.208 | 0.252 | 0.240 | 0.271 | 0.225 | 0.259 | 0.206 | 0.277 | 0.242 | 0.298 | 0.219 | 0.261 | 0.240 | 0.301 | 0.237 | 0.296 |
| | 336 | **0.260** | **0.281** | 0.263 | 0.293 | 0.278 | 0.296 | 0.263 | 0.293 | 0.292 | 0.307 | 0.278 | 0.297 | 0.272 | 0.335 | 0.287 | 0.335 | 0.280 | 0.306 | 0.284 | 0.334 | 0.283 | 0.335 |
| | 720 | **0.339** | **0.334** | 0.341 | 0.343 | 0.358 | 0.349 | 0.345 | 0.345 | 0.364 | 0.353 | 0.354 | 0.348 | 0.398 | 0.418 | 0.351 | 0.386 | 0.365 | 0.359 | 0.351 | 0.387 | 0.345 | 0.381 |
| | Avg | **0.237** | **0.259** | 0.240 | 0.271 | 0.258 | 0.279 | 0.245 | 0.274 | 0.272 | 0.291 | 0.259 | 0.281 | 0.259 | 0.315 | 0.271 | 0.320 | 0.259 | 0.287 | 0.267 | 0.318 | 0.265 | 0.317 |
| Traffic | 96 | 0.414 | **0.251** | 0.550 | 0.355 | **0.395** | 0.268 | 0.473 | 0.287 | 0.649 | 0.389 | 0.544 | 0.359 | 0.522 | 0.290 | 0.805 | 0.493 | 0.593 | 0.321 | 0.521 | 0.310 | 0.650 | 0.396 |
| | 192 | **0.432** | **0.257** | 0.527 | 0.337 | 0.417 | 0.276 | 0.486 | 0.294 | 0.601 | 0.366 | 0.540 | 0.354 | 0.530 | 0.293 | 0.756 | 0.474 | 0.617 | 0.336 | 0.536 | 0.314 | 0.598 | 0.370 |
| | 336 | **0.446** | **0.265** | 0.537 | 0.342 | 0.433 | 0.283 | 0.488 | 0.298 | 0.609 | 0.369 | 0.551 | 0.358 | 0.558 | 0.305 | 0.762 | 0.477 | 0.629 | 0.336 | 0.550 | 0.321 | 0.605 | 0.373 |
| | 720 | **0.485** | **0.286** | 0.570 | 0.359 | 0.467 | 0.302 | 0.536 | 0.314 | 0.647 | 0.387 | 0.586 | 0.375 | 0.589 | 0.328 | 0.719 | 0.449 | 0.640 | 0.350 | 0.571 | 0.329 | 0.645 | 0.394 |
| | Avg | **0.444** | **0.265** | 0.546 | 0.348 | 0.428 | 0.282 | 0.496 | 0.298 | 0.626 | 0.378 | 0.555 | 0.362 | 0.550 | 0.304 | 0.760 | 0.473 | 0.620 | 0.336 | 0.544 | 0.319 | 0.625 | 0.383 |
| 1$^{st}$Count | | 30 | 35 | 1 | 0 | 5 | 0 | 0 | 0 | 0 | 0 | 0 | 0 | 0 | 0 | 0 | 0 | 0 | 0 | 0 | 0 | 0 | 0 |

Table 13: Full results of ablation studies for ETTm1, ETTm2, ETTh1, and ETTh2. We fix the look-back window $L = 96$ and make predictions for $T = \{96, 192, 336, 720\}$. The best is highlighted in **red** and the runner-up in blue.

| GDD MLP | Channel Mixup | Metric | ETTm1 96 | 192 | 336 | 720 | ETTm2 96 | 192 | 336 | 720 | ETTh1 96 | 192 | 336 | 720 | ETTh2 96 | 192 | 336 | 720 |
|---|---|---|---|---|---|---|---|---|---|---|---|---|---|---|---|---|---|---|
| - | - | MSE | 0.320 | 0.373 | 0.401 | 0.455 | 0.175 | 0.244 | 0.304 | 0.404 | 0.374 | **0.422** | **0.458** | 0.471 | 0.284 | **0.361** | **0.410** | **0.415** |
| | | MAE | 0.345 | 0.373 | 0.398 | 0.434 | 0.254 | 0.300 | 0.339 | 0.397 | 0.390 | 0.418 | 0.439 | 0.463 | 0.331 | **0.380** | 0.419 | 0.435 |
| - | ✓ | MSE | 0.312 | 0.370 | 0.399 | 0.453 | 0.177 | 0.242 | 0.301 | 0.402 | **0.372** | 0.423 | 0.462 | **0.462** | 0.284 | 0.363 | 0.428 | 0.419 |
| | | MAE | 0.339 | 0.369 | 0.390 | 0.429 | 0.255 | 0.297 | 0.337 | 0.396 | 0.390 | 0.419 | 0.437 | 0.459 | 0.331 | **0.380** | 0.427 | **0.435** |
| ✓ | - | MSE | 0.317 | 0.363 | 0.391 | 0.481 | 0.172 | 0.238 | **0.293** | 0.397 | 0.382 | 0.430 | 0.470 | 0.465 | 0.285 | 0.364 | 0.417 | 0.423 |
| | | MAE | 0.341 | 0.372 | 0.394 | 0.441 | 0.250 | 0.294 | **0.333** | 0.393 | 0.394 | 0.420 | 0.440 | **0.457** | 0.332 | 0.382 | 0.422 | 0.439 |
| ✓ | ✓ | MSE | **0.308** | **0.359** | **0.390** | **0.447** | **0.171** | **0.235** | 0.296 | **0.392** | 0.372 | 0.422 | 0.466 | 0.470 | **0.281** | **0.361** | 0.413 | 0.419 |
| | | MAE | **0.338** | **0.364** | **0.389** | **0.425** | **0.248** | **0.292** | 0.334 | **0.391** | 0.386 | 0.416 | 0.438 | 0.461 | **0.329** | 0.381 | 0.419 | 0.435 |

Table 14: Full results of ablation studies for Electricity, Weather, and Traffic. We fix the look-back window $L = 96$ and make predictions for $T = \{96, 192, 336, 720\}$. The best is highlighted in **red** and the runner-up in blue.

| GDD MLP | Channel Mixup | Metric | Electricity 96 | 192 | 336 | 720 | Weather 96 | 192 | 336 | 720 | Traffic 96 | 192 | 336 | 720 |
|---|---|---|---|---|---|---|---|---|---|---|---|---|---|---|
| - | - | MSE | 0.163 | 0.173 | 0.190 | 0.233 | 0.173 | 0.219 | 0.276 | 0.352 | 0.454 | 0.467 | 0.479 | 0.514 |
| | | MAE | 0.243 | 0.253 | 0.270 | 0.306 | 0.207 | 0.247 | 0.289 | 0.339 | **0.251** | **0.257** | **0.264** | **0.284** |
| - | ✓ | MSE | 0.164 | 0.173 | 0.190 | 0.229 | 0.173 | 0.218 | 0.273 | 0.352 | 0.467 | 0.463 | 0.483 | 0.518 |
| | | MAE | 0.243 | 0.252 | 0.270 | 0.302 | 0.206 | 0.246 | 0.287 | 0.339 | 0.257 | 0.259 | 0.268 | 0.285 |
| ✓ | - | MSE | 0.147 | 0.166 | **0.174** | 0.212 | **0.150** | **0.199** | 0.269 | 0.345 | 0.471 | 0.499 | 0.546 | 0.585 |
| | | MAE | 0.240 | 0.258 | 0.265 | 0.299 | **0.186** | **0.236** | 0.287 | 0.337 | 0.272 | 0.281 | 0.289 | 0.300 |
| ✓ | ✓ | MSE | **0.141** | **0.157** | 0.175 | **0.203** | **0.150** | 0.200 | 0.260 | 0.339 | 0.414 | 0.432 | 0.446 | 0.485 |
| | | MAE | **0.231** | **0.245** | **0.265** | **0.289** | 0.187 | 0.236 | 0.281 | 0.334 | 0.251 | 0.257 | 0.265 | 0.286 |

Table 15: Full results of the long-term forecasting task with SSM variants. We fix the look-back window $L = 96$ and make predictions for $T = \{96, 192, 336, 720\}$. Avg means the average metrics for four prediction lengths. The best is highlighted in red and the runner-up in blue.

| Models | | CMamba (Ours) | | Bi-Mamba+ (2024) | | SiMBA (2024) | | Time-SSM (2024) | | SAMBA (2024) | | S-Mamba (2024d) | |
|---|---|---|---|---|---|---|---|---|---|---|---|---|---|
| Metric | | MSE | MAE | MSE | MAE | MSE | MAE | MSE | MAE | MSE | MAE | MSE | MAE |
| ETTm1 | 96 | 0.308 | 0.338 | 0.320 | 0.360 | 0.324 | 0.360 | 0.329 | 0.365 | 0.315 | 0.357 | 0.331 | 0.368 |
| | 192 | 0.359 | 0.364 | 0.361 | 0.383 | 0.363 | 0.382 | 0.370 | 0.379 | 0.360 | 0.383 | 0.371 | 0.387 |
| | 336 | 0.390 | 0.389 | 0.386 | 0.402 | 0.395 | 0.405 | 0.396 | 0.402 | 0.389 | 0.405 | 0.417 | 0.418 |
| | 720 | 0.447 | 0.425 | 0.445 | 0.437 | 0.451 | 0.437 | 0.449 | 0.440 | 0.448 | 0.440 | 0.471 | 0.448 |
| | Avg | 0.376 | 0.379 | 0.378 | 0.396 | 0.383 | 0.396 | 0.386 | 0.397 | 0.378 | 0.396 | 0.398 | 0.405 |
| ETTm2 | 96 | 0.171 | 0.248 | 0.176 | 0.263 | 0.177 | 0.263 | 0.176 | 0.260 | 0.172 | 0.259 | 0.179 | 0.263 |
| | 192 | 0.235 | 0.292 | 0.242 | 0.304 | 0.245 | 0.306 | 0.246 | 0.305 | 0.238 | 0.301 | 0.253 | 0.310 |
| | 336 | 0.296 | 0.334 | 0.304 | 0.344 | 0.304 | 0.343 | 0.305 | 0.344 | 0.300 | 0.340 | 0.312 | 0.348 |
| | 720 | 0.392 | 0.391 | 0.402 | 0.402 | 0.400 | 0.399 | 0.406 | 0.405 | 0.394 | 0.394 | 0.412 | 0.408 |
| | Avg | 0.273 | 0.316 | 0.281 | 0.328 | 0.282 | 0.328 | 0.283 | 0.329 | 0.276 | 0.324 | 0.289 | 0.332 |
| ETTh1 | 96 | 0.372 | 0.386 | 0.378 | 0.395 | 0.379 | 0.395 | 0.377 | 0.394 | 0.376 | 0.400 | 0.386 | 0.406 |
| | 192 | 0.422 | 0.416 | 0.427 | 0.428 | 0.432 | 0.424 | 0.423 | 0.424 | 0.432 | 0.429 | 0.448 | 0.444 |
| | 336 | 0.466 | 0.438 | 0.471 | 0.445 | 0.473 | 0.443 | 0.466 | 0.437 | 0.477 | 0.437 | 0.494 | 0.468 |
| | 720 | 0.470 | 0.461 | 0.470 | 0.457 | 0.483 | 0.469 | 0.452 | 0.448 | 0.488 | 0.471 | 0.493 | 0.488 |
| | Avg | 0.433 | 0.425 | 0.437 | 0.431 | 0.442 | 0.433 | 0.430 | 0.426 | 0.443 | 0.434 | 0.455 | 0.452 |
| ETTh2 | 96 | 0.281 | 0.329 | 0.291 | 0.342 | 0.290 | 0.339 | 0.290 | 0.341 | 0.288 | 0.340 | 0.298 | 0.349 |
| | 192 | 0.361 | 0.381 | 0.368 | 0.392 | 0.373 | 0.390 | 0.368 | 0.387 | 0.373 | 0.390 | 0.379 | 0.398 |
| | 336 | 0.413 | 0.419 | 0.407 | 0.424 | 0.376 | 0.406 | 0.416 | 0.430 | 0.380 | 0.406 | 0.417 | 0.432 |
| | 720 | 0.419 | 0.435 | 0.421 | 0.439 | 0.407 | 0.431 | 0.424 | 0.439 | 0.412 | 0.432 | 0.431 | 0.449 |
| | Avg | 0.368 | 0.391 | 0.372 | 0.399 | 0.362 | 0.392 | 0.375 | 0.399 | 0.363 | 0.392 | 0.381 | 0.407 |
| Electricity | 96 | 0.141 | 0.231 | 0.140 | 0.238 | 0.165 | 0.253 | - | - | 0.146 | 0.244 | 0.142 | 0.238 |
| | 192 | 0.157 | 0.245 | 0.155 | 0.253 | 0.173 | 0.262 | - | - | 0.164 | 0.260 | 0.169 | 0.267 |
| | 336 | 0.175 | 0.265 | 0.170 | 0.269 | 0.188 | 0.277 | - | - | 0.179 | 0.274 | 0.178 | 0.275 |
| | 720 | 0.203 | 0.289 | 0.197 | 0.293 | 0.214 | 0.305 | - | - | 0.206 | 0.300 | 0.207 | 0.303 |
| | Avg | 0.169 | 0.258 | 0.166 | 0.263 | 0.185 | 0.274 | - | - | 0.174 | 0.270 | 0.174 | 0.271 |
| Weather | 96 | 0.150 | 0.187 | 0.159 | 0.205 | 0.176 | 0.219 | 0.167 | 0.212 | 0.165 | 0.211 | 0.166 | 0.210 |
| | 192 | 0.200 | 0.236 | 0.205 | 0.249 | 0.222 | 0.260 | 0.217 | 0.255 | 0.214 | 0.255 | 0.215 | 0.253 |
| | 336 | 0.260 | 0.281 | 0.264 | 0.291 | 0.275 | 0.297 | 0.274 | 0.291 | 0.271 | 0.297 | 0.276 | 0.298 |
| | 720 | 0.339 | 0.334 | 0.343 | 0.344 | 0.350 | 0.349 | 0.351 | 0.345 | 0.346 | 0.347 | 0.353 | 0.349 |
| | Avg | 0.237 | 0.259 | 0.243 | 0.272 | 0.256 | 0.281 | 0.252 | 0.277 | 0.249 | 0.278 | 0.253 | 0.278 |
| Traffic | 96 | 0.414 | 0.251 | 0.375 | 0.258 | 0.468 | 0.268 | - | - | 0.403 | 0.270 | 0.381 | 0.261 |
| | 192 | 0.432 | 0.257 | 0.394 | 0.269 | 0.413 | 0.317 | - | - | 0.427 | 0.278 | 0.397 | 0.267 |
| | 336 | 0.446 | 0.265 | 0.406 | 0.274 | 0.529 | 0.284 | - | - | 0.440 | 0.284 | 0.423 | 0.276 |
| | 720 | 0.485 | 0.286 | 0.440 | 0.288 | 0.564 | 0.297 | - | - | 0.470 | 0.302 | 0.458 | 0.300 |
| | Avg | 0.444 | 0.265 | 0.404 | 0.272 | 0.494 | 0.292 | - | - | 0.435 | 0.284 | 0.415 | 0.276 |
| 1st Count | | 18 | 31 | 12 | 0 | 3 | 2 | 3 | 2 | 0 | 2 | 0 | 0 |

## D FULL RESULTS

### D.1 FULL MAIN RESULTS

Here, we present the complete results of all chosen models and our CMamba under four different prediction lengths in Table 12. Generally, the proposed CMamba demonstrates stable performance across various datasets and prediction lengths, consistently ranking among the top performers. Specifically, our model ranks top 1 in **65** out of 70 settings and ranks top 2 in **all** settings, while the runner-up, iTransformer (Liu et al., 2023a) ranks top 1 in only 5 settings and top 2 in 18 settings.

### D.2 FULL ABLATION RESULTS OF GDD-MLP AND CHANNEL MIXUP

In the main text, we only present the improvements brought by the proposed modules in the average case. To validate the effectiveness of our design, we provide the complete results in Table 13 and Table 14. Consistent with our claims, GDD-MLP alone can easily lead to oversmoothing. However, when combined with the Channel Mixup, our model consistently achieves state-of-the-art performance.

### D.3 COMPARISON WITH OTHER SSMS

Since Mamba (Gu & Dao, 2023) was proposed, there have been many remarkable works trying to apply it to multivariate time series prediction, e.g., Bi-Mamba+ (Liang et al., 2024), SiMBA (Patro & Agneeswaran, 2024), Time-SSM (Hu et al., 2024), SAMBA (Weng et al., 2024), and S-Mamba (Wang et al., 2024d). In Table 15, we compare our CMamba with these methods on the same 7 real-world datasets. It is worth noting that since some methods are tested on other datasets in the original paper, there will be missing results on some of the datasets we use. Among all these Mamba variants, our model achieves the best average performance. Specifically, our model ranks top 1 in **49** out of 70 settings and ranks top 2 in **59** settings, while the runner-up, Bi-Mamba+ ranks top 1 in only 12 settings and top 2 in 39 settings. The results demonstrate the superiority of our model.

## E SHOWCASES

### E.1 COMPARISON WITH BASELINES

As depicted in Fig. 10, Fig. 11, Fig. 12, and Fig. 13, Fig. 14, Fig. 15, we visualize the forecasting results on the Electricity and Traffic dataset of our model, ModernTCN (Luo & Wang, 2024), and TimeMixer (Wang et al., 2023). Overall, our model fits the data better. Especially when dealing with non-periodic changes. For instance, in Prediction-96 of the Electricity dataset, our model exhibits significantly better performance compared to the others.

### E.2 MORE SHOWCASES

As shown in Fig. 16, Fig. 17, Fig. 18, Fig. 19, and Fig. 20, we visualize the forecasting results of other datasets under CMamba. The results demonstrate that CMamba achieves consistently stable performance under various datasets.

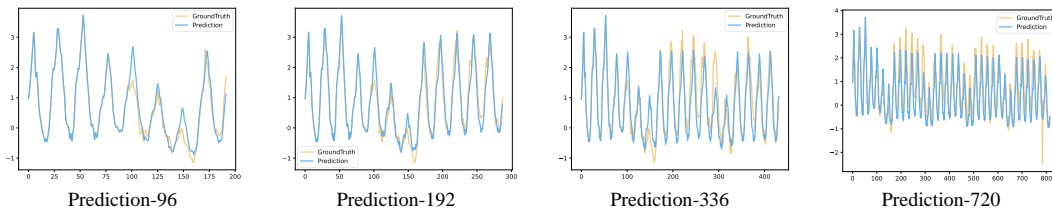

Figure 10: Prediction cases for Electricity under CMamba.

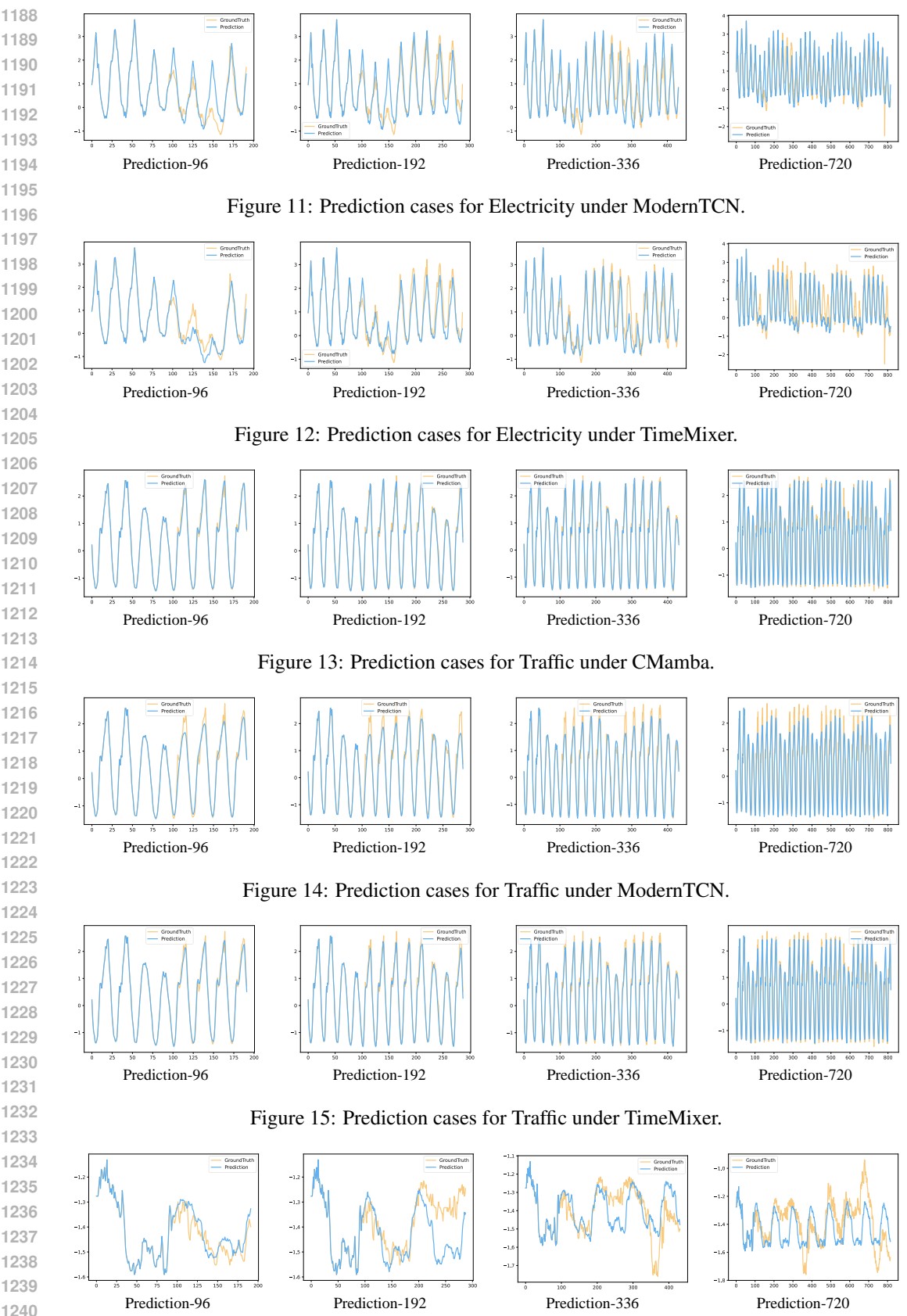

Figure 11: Prediction cases for Electricity under ModernTCN.

Figure 12: Prediction cases for Electricity under TimeMixer.

Figure 13: Prediction cases for Traffic under CMamba.

Figure 14: Prediction cases for Traffic under ModernTCN.

Figure 15: Prediction cases for Traffic under TimeMixer.

Figure 16: Prediction cases for ETTm1 under CMamba.

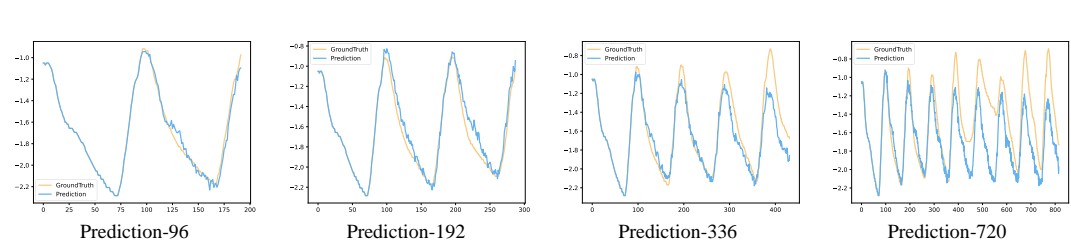

Figure 17: Prediction cases for ETTm2 under CMamba.

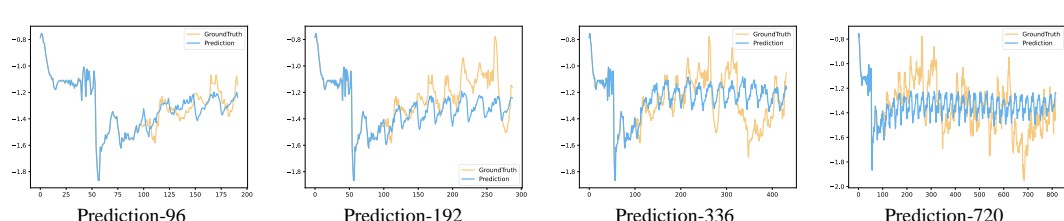

Figure 18: Prediction cases for ETTh1 under CMamba.

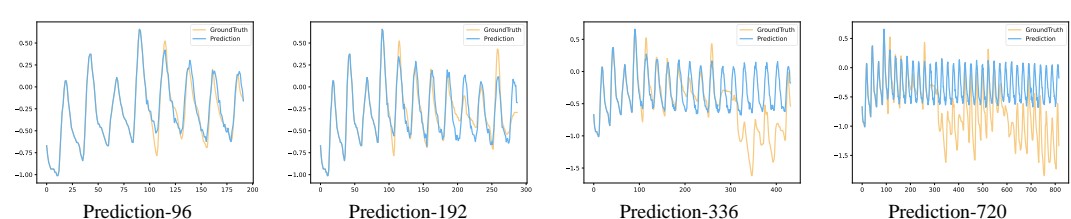

Figure 19: Prediction cases for ETTh2 under CMamba.

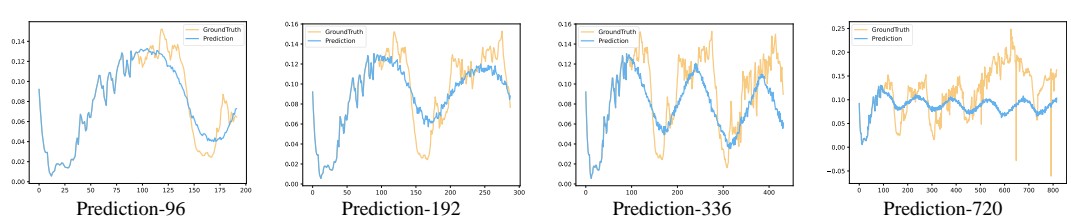

Figure 20: Prediction cases for Weather under CMamba.

