# OpenReview forum: "CMamba: Channel Correlation Enhanced State Space Models for Multivariate Time Series Forecasting"
_ICLR.cc/2025/Conference — Submitted to ICLR 2025_

### Official Review · Reviewer_Wmo3 · 2024-10-15

**Soundness:** 3
**Presentation:** 3
**Contribution:** 3
**Rating:** 6
**Confidence:** 3

**Summary:**

The paper titled "CMamba: Channel Correlation Enhanced State Space Model for Multivariate Time Series Prediction" introduces a new model called CMamba, which builds upon the existing Mamba state space model and addresses the key limitation of capturing cross channel dependencies that are crucial for multivariate time series prediction (MTSF). The CMamba architecture integrates two main components: (1) the M-Mamba module for modeling temporal dependencies, and (2) the Global Data Dependency MLP (GDD-MLP) for capturing cross channel correlations. In addition, a channel mixing strategy was introduced to improve generalization ability and alleviate overfitting. The author validated their method through extensive experiments on seven benchmark datasets, demonstrating that CMamba outperforms state-of-the-art models in long-term prediction tasks.

**Strengths:**

Originality: M-Mamba is used for time-dependent analysis, while GDD-MLP is used for cross channel correlation, which creatively enhances the state space model and solves the limitations of existing MTSF methods.
Quality: This article presents a set of well executed experiments that validate the effectiveness of CMamba in multiple datasets and prediction tasks. The consistent and statistically significant results demonstrate the robustness of the proposed model.
Clarity: The model architecture and experimental setup are clearly described and accompanied by useful charts (as shown in Figure 3). Tables and charts are well organized for easy interpretation of results.
Meaning: CMamba solves an important problem in time series prediction - modeling cross channel dependencies - which has implications for many real-world applications such as traffic management and energy consumption prediction.

**Weaknesses:**

Long term dependency exploration: Although the GDD-MLP module has shown promising results in mid-term time series prediction, its effectiveness in handling very long-term dependencies (such as prediction ranges exceeding 720) has not been fully explored. Additional experiments on long-term prediction tasks can further validate the robustness of the model.
Generalization in different fields: This article mainly focuses on standard datasets such as electricity and transportation, but it is beneficial to test CMamba's generalization ability on other types of time series data (such as financial or biological datasets), which may have different time and cross channel characteristics.

**Questions:**

Has the model been tested on datasets outside the field explored in this article, such as financial or biological time series? Understanding the generalization level of CMamba to different types of multivariate time series would be valuable.
Can the author provide more insights into the computational efficiency of CMamba compared to Transformer based models such as iTransformer? A more detailed analysis of computational trade-offs may be helpful for practitioners considering large-scale deployment of the model.

---

> ### Author Response · Authors · 2024-11-16
> **Response to Reviewer Wmo3**
>
> We sincerely thank Reviewer Wmo3 for providing thorough and insightful comments. Here are our responses to your concerns and questions.
>
> - **W1: Lack of experiments under very long-term horizon (longer than 720).**
>
> Thanks for your insightful suggestions. We follow the common setting of long-term multivariate time series forecasting, i.e., predicting the future for **96, 192, 336, and 720** steps, so we do not consider other prediction lengths in our paper. However, it is important and meaningful to make ultra-long-term forecasts. Hence, we add experiments under **960** future steps in the following table, where CMamba ranks top 1 in **10 out of 14** settings:
>
> ||ETTm1||ETTm2||ETTh1||ETTh2||Weather||Electricity||Traffic||
> |:-:|:-:|:-:|:-:|:-:|:-:|:-:|:-:|:-:|:-:|:-:|:-:|:-:|:-:|:-:|
> ||MSE|MAE|MSE|MAE|MSE|MAE|MSE|MAE|MSE|MAE|MSE|MAE|MSE|MAE|
> |CMamba|0.483|**0.446**|**0.433**|**0.416**|0.535|0.499|**0.443**|**0.455**|**0.364**|**0.354**|**0.217**|**0.301**|0.495|**0.289**|
> |ModernTCN|0.485|0.455|0.442|0.421|0.534|0.499|0.462|0.468|**0.364**|0.361|0.262|0.335|0.581|0.366|
> |iTransformer|0.519|0.475|0.452|0.430|**0.525**|0.499|0.475|0.478|0.385|0.371|0.261|0.338|0.473|0.309|
> |TimeMixer|0.483|0.458|0.437|0.423|0.550|0.504|0.496|0.493|0.371|0.365|0.250|0.333|0.556|0.327|
> |PatchTST|**0.482**|0.459|0.435|0.420|0.580|0.528|0.456|0.466|0.380|0.366|0.252|0.333|0.526|0.329|
> |RLinear|0.513|0.465|0.448|0.423|0.529|**0.497**|0.463|0.468|0.389|0.372|0.278|0.343|0.651|0.388|
>
> The above results demonstrate the powerful ultra-long time series prediction capability of CMamba, which further proves the effectiveness of our modeling of cross-time dependencies and cross-channel dependencies.
>
> - **W2 & Q1: Generalization of CMamba in other fields (like financial or biological datasets).**
>
> Thanks for your valuable suggestions. We have added experiments on the **Exchange** (the daily exchange rates of eight different countries ranging from 1990 to 2016) and **ILI** (the weekly recorded influenza-like illness (ILI) patients data in the United States between 2002 and 2021) datasets. The look-back length is 96 and the horizons are set to the commonly used lengths (96, 192, 336, 720 for **Exchange** and 24, 36, 48, 60 for **ILI**). The results show that our model achieves the best average performance, which further proves the versatility of Cmamba.
>
> |||CMamba||ModernTCN||iTransformer||TimeMixer||PatchTST||RLinear||DLinear||
> |:-:|:-:|:-:|:-:|:-:|:-:|:-:|:-:|:-:|:-:|:-:|:-:|:-:|:-:|:-:|:-:|
> ||T|MSE|MAE|MSE|MAE|MSE|MAE|MSE|MAE|MSE|MAE|MSE|MAE|MSE|MAE|
> |Exchange|96|**0.082**|**0.198**|0.103|0.228|0.086|0.206|*0.085*|*0.203*|0.088|0.205|0.093|0.217|0.088|0.218|
> ||192|**0.174**|**0.295**|0.200|0.319|0.177|*0.299*|0.177|*0.299*|*0.176*|*0.299*|0.184|0.307|*0.176*|0.315|
> ||336|0.332|*0.415*|0.354|0.432|*0.331*|0.417|0.357|0.434|**0.301**|**0.397**|0.351|0.432|0.313|0.427|
> ||720|0.872|0.704|0.892|0.717|*0.847*|**0.691**|1.077|0.765|0.901|0.714|0.886|0.714|**0.839**|*0.695*|
> ||Avg|*0.365*|**0.403**|0.387|0.424|**0.360**|**0.403**|0.424|0.425|0.367|*0.404*|0.378|0.417|0.354|0.414|
> |ILI|24|**1.832**|**0.826**|*1.953*|0.906|2.293|1.032|2.021|0.942|1.997|*0.885*|2.255|1.036|2.408|1.102|
> ||36|*1.747*|**0.844**|**1.659**|0.885|2.235|1.003|1.905|0.919|1.852|*0.880*|2.053|0.969|2.332|1.090|
> ||48|**1.783**|**0.836**|1.986|0.942|2.129|1.003|1.870|0.910|*1.866*|*0.890*|2.058|0.989|2.384|1.122|
> ||60|*1.804*|**0.871**|2.152|1.025|2.088|1.013|1.961|0.944|**1.785**|*0.879*|2.002|0.974|2.338|1.085|
> ||Avg|**1.792**|**0.844**|1.937|0.940|2.186|1.013|1.939|0.929|*1.875*|*0.884*|2.092|0.992|2.365|1.100|
> |$1^{st}$ count||**5**|**8**|1|0|1|*2*|0|0|*2*|1|0|0|1|0|
>
> - **Q2: Computational analysis**
>
> Thanks for your suggestions. We have added the analysis of model efficiency, including model size and training time, in **Appendix C.2 Efficiency Analysis** in the new updated version. Here we show the results on the Weather dataset, with more details presented in **Appendix C.2**.
>
> |Model|RLinear|TimeMixer|TimesNets|CMamba|ModernTCN|iTransformer|PatchTST|
> |:-:|:-:|:-:|:-:|:-:|:-:|:-:|:-:|
> |Parameters|69.84k|0.22M|1.25M|**1.35M**|3.09M|5.15M|10.74M|
> |TrainingTime (ms/iter)|13|43|325|**29**|29|28|33|
> |MSE|0.364|0.345|0.365|**0.339**|0.343|0.358|0.354|
>
> We can find that CMamba is the most lightweight model on average except for the Linear-based model (e.g., RLinear and TimeMixer). This indicates that CMamba’s excellent performance stems more from its capacity to capture historical time series patterns than from an increase in parameters. However, CMamba's running speed is not outstanding, which is primarily due to the suboptimal parallelism inherent to state space models (SSMs) compared to attention-based architectures. Nevertheless, CMamba's strong performance and lightweight model size still make it a highly efficient model, effectively balancing accuracy and computational efficiency.

---

### Official Review · Reviewer_ah3a · 2024-10-28

**Soundness:** 2
**Presentation:** 3
**Contribution:** 2
**Rating:** 5
**Confidence:** 5

**Summary:**

The article presents two main contributions: M-Mamba and GDD-MLP+Channel Mixup. The former is a further exploration of the application of Mamba in time-series tasks, but in my opinion, there is a certain degree of controversy. The latter is a good solution to the problem of time-series channel dependence, with sufficient argumentation and experiments. The overall performance of the model is good, but it mainly comes from the latter contribution.

**Strengths:**

The overall writing logic of the article is clear, and the drawings are relatively clear. The results are presented in detail.
I think the combination of GDD-MLP+Channel mixup is a good contribution point. Moreover, applying it to other frameworks further proves its effectiveness, and an analysis of its efficiency is also conducted. However, on the contrary, as can be seen from Table 3, M-Mamba has no advantage compared to the state-of-the-art.

**Weaknesses:**

1. Line 53-55, There have been many studies on the application of past state space models in temporal dependency. Including: S4, SpaceTime, and the direct application of Mamba itself in temporal tasks has long existed, including the temporal dependency based on patches same with your M-mamba. I think it is worthy of research to consider cross-channel on the basis of temporal for mamba-based models. However, the problem raised here is not very accurate and the theme is not very clear.
2. Line 61-64, MLP is a multi-layer perceptron. In the Transformer, FFN is a fully connected mapping layer. It is somewhat inappropriate to say that it is part of the Transformer. I don't know what is the meaning of expressing FFN as a part of Attention? Because in iTransformer, the overall method of Attention is applied to channel dependence. The function of FFN in the attention mechanism has no direct connection with the dependence between channels. There is no relevant content in iTransformer.
3. Line 84-91 This paragraph in the introduction does not meet the basic requirements of paper writing very well. The experimental results proving the effectiveness of the method should be placed in the experimental part.
4. Line 132-140, you also mentioned S4, and later in Table 12 also compared the effect with the mamba-based model. Still the above problem, there exist studies on SSM in temporal. It should still highlight that the model considers cross-channel on the basis of temporal, rather than generally saying that there is insufficient research on temporal.
5. For M-mamba (removing convolution and using FI), i am not sure that it is a general and effective improvement. I think there is controversy as a contribution. Mainly because the ablation experiment in section 5.2 is only carried out on one dataset, Weather, and the performance difference is not significant. In addition to the experiment, there is a lack of necessary explanations. For example, In Time-SSM Table 2 gives a detailed comparison with the patch Mamba (Mamba4TS).

**Questions:**

Line 64-68, The view put forward is not reasonable in my opinion. Why should MLP be an ideal structure? There is no basis. Can you provide relevant paper citation basis? Has there been any work in the past using MLP to practice between variables or similar work?

---

> ### Author Response · Authors · 2024-11-16
> **Response to Reviewer ah3a (Part 1)**
>
> We sincerely thank Reviewer ah3a for providing thorough and insightful comments. Here are our responses to your concerns and questions.
>
> - **W1: Inaccurate description of the current status and problems of Mamba-based research in time series.**
>
> Thanks for your valuable suggestions. We have elaborated on the current status and existing problems of Mamba in time series in the **latest uploaded version**, where **Lines 53-59** describe the shortcomings of existing Mamba research in the time series and our motivation:
> 1. We point out that some current Mamba-based models in time series, e.g., S-Mamba, SiMBA, focus on replacing attention with Mamba, but ignore the study of Mamba's components.
> 2. We point out that Mamba lacks cross-channel dependency modeling capabilities. The current approach of using the SSM kernel to model channel dependencies is too complex and ignores the fact that channels are not sequences.
>
> Hopefully, this will clear up your concerns.
>
> - **W2: The description of the Transformer in Lines 61-64 is ambiguous.**
>
> Thanks for your careful reading. The Transformer we are talking about actually refers to the entire **Transformer encoder block**, including the **multihead self-attention module and the FFN module**. Therefore, the MLP we mentioned in our paper refers to the FFN module. We are sorry for the ambiguity. We have corrected this in the **latest uploaded version (Line 65-69)**.
>
> - **W3: Experimental results should not be included in the introduction.**
>
> Thanks for your suggestions. The Fig. 2 in the introduction section, which exhibits some experimental results is removed in the **newly updated version**.
>
> - **W4: The description of the current status of Mamba in time series in related work is too general.**
>
> Thanks for the valuable suggestions. We have elaborated the **Section 2.1** in the latest uploaded version (**Line 127-139**):
> 1. We have cited and elaborated on more studies about state space models.
> 2. We introduce the latest state-space models for time series, e.g., S-Mamba, SiMBA, Time-SSM, and Bi-Mamba+. We describe their methods and point out their shortcomings.
> 3. We emphasize that our research focuses on studying the effectiveness of Mamba components and modeling cross-channel dependencies.
>
> Hopefully, this will clear up your concerns.

---

> > ### Comment · Reviewer_ah3a · 2024-11-25
> >
> > It has been changed to FFN here. But isn't the approach in the subsequent text still MLP? The explanation in this paragraph doesn't match the subsequent text at all. It's not just a matter of changing words.

---

> > > ### Author Response · Authors · 2024-11-25
> > >
> > > **The transformer encoder consists of the multihead self-attention layer and FFN (MLP) layer**. iTransformer captures cross-channel dependencies with self-attention and cross-time dependencies with FFN (MLP). However, FFN (MLP) does not work well in capturing cross-channel dependencies. We reflected on the reasons for its failure and proposed GDD-MLP and channel mixup. We think we have made it clear.

---

> > > > ### Comment · Reviewer_ah3a · 2024-11-27
> > > >
> > > > I think you are familiar with linear models, such as DLinear and RLinear. They utilize Feed-Forward Networks (FFN) instead of nonlinear Multi-Layer Perceptrons (MLP), and there are no activation functions! Do you think MLP and FFN are the same thing? The iTransformer mentions that FFN is used to capture temporal dependencies, but it doesn't say anything related to MLP. Please understand the models objectively, and think about my questions carefully before answering. I recognize the effectiveness of GDD-MLP. I already emphasized this during the review. However, I completely disagree with the problems that GDD-MLP aims to solve and the underlying logic. Given these obvious logical errors, I will change the score to 3.

---

> > > > > ### Author Response · Authors · 2024-11-27
> > > > >
> > > > > Thank you for your reply again. We think our main disagreement lies in the difference between FFN and MLP. We apologize for our lack of rigor.
> > > > >
> > > > > Although the iTransformer paper mentions FFN, its code implementation (including the example in Figure 4c of the original paper) is two dense layers with activation. The MLP we refer to also has two dense layers with activation.

---

> > > > > > ### Comment · Reviewer_ah3a · 2024-11-27
> > > > > >
> > > > > > No, I should apologize. You are right. I read the code and paper of iTransformer and reviewed CMamba again. I recognize that GDD-MLP is a good contribution. However, the overall contributions of the article are slightly lacking. You also applied GDD-MLP to other models to demonstrate its universality, which makes the research and improvement on Mamba seem even less significant. Still, I didn't see any improvement in M-Mamba compared with PatchMamba (without no GDD-MLP), the updated experiment in Appendix C.1 is supplementary evidence, but not the fundamental one. Do you agree with the logic of the ablation settings for M-Mamba? I'll keep the score at 5.

---

> ### Author Response · Authors · 2024-11-16
> **Response to Reviewer ah3a (Part 2)**
>
> - **W5: Ablation study for M-Mamba is not convincing.**
>
> Due to space limitations, our discussion in M-Mamba is indeed insufficient. We have added the ablation results of vanilla Mamba and M-Mamba in the newly updated version (**Appendix C.1**). Here we briefly explain its content. As shown in the following table, where **bold** means improved performance (we only modify the backbone, so GDD-MLP and Channel Mixup are still adopted).
>
> || | ETTm1 || ETTm2 || ETTh1 || ETTh2 || Electricity || Weather || Traffic||
> |:-:|:-:|:-:|:-:|:-:|:-:|:-:|:-:|:-:|:-:|:-:|:-:|:-:|:-:|:-:|:-:|
> || | MSE | MAE  | MSE | MAE | MSE | MAE | MSE | MAE | MSE | MAE | MSE | MAE | MSE | MAE|
> |Mamba| 96| 0.312 | 0.341 | 0.173 | 0.251 | 0.376 | 0.390 | 0.283 | 0.331 | 0.140 | 0.231 | 0.152 | 0.189 | 0.404 | 0.250|
> || 192 | 0.365 | 0.367 | 0.239 | 0.296 | 0.423 | 0.417 | 0.360 | 0.381 | 0.159 | 0.247 | 0.201 | 0.236 | 0.429 | 0.255|
> || 336 | 0.398 | 0.392 | 0.300 | 0.336 | 0.468 | 0.439 | 0.416 | 0.422 | 0.177 | 0.265 | 0.264 | 0.283 | 0.455 | 0.265|
> || 720 | 0.455 | 0.427 | 0.398 | 0.393 | 0.474 | 0.466 | 0.419 | 0.437 | 0.214 | 0.298 | 0.344 | 0.337 | 0.483 | 0.286|
> || Avg | 0.383 | 0.381 | 0.277 | 0.319 | 0.435 | 0.428 | 0.370 | 0.393 | 0.173 | 0.260  | 0.240 | 0.261 | 0.443 | 0.264|
> |M-Mamba| 96| **0.308** | **0.338** | **0.171** | **0.248** | **0.372** | **0.386** | **0.281** | **0.329** | 0.141 | 0.231 | **0.150** | **0.187** | 0.414 | 0.251|
> || 192 | **0.359** | **0.364** | **0.235** | **0.292** | **0.422** | **0.416** | 0.361 | 0.381 | **0.157** | **0.245** | **0.200** | **0.236** | 0.432 | 0.257|
> || 336 | **0.390** | **0.389** | **0.296** | **0.334** | **0.466** | **0.438** | **0.413** | **0.419** | **0.175** | **0.265** | **0.260** | **0.281** | **0.446** | **0.265**|
> || 720 | **0.447** | **0.425** | **0.392** | **0.391** | **0.470** | **0.461** | **0.419** | **0.435** | **0.203** | **0.289** | **0.339** | **0.334** | 0.485 | 0.286|
> || Avg | **0.376** | **0.379** | **0.273** | **0.316** | **0.433** | **0.425** | **0.368** | **0.391** | **0.169** | **0.258** | **0.237** | **0.259** | 0.444 | 0.265|
> ||Promotion Count | 5 / 5 | 5 / 5 | 5 / 5 | 5 / 5 | 5 / 5 | 5 / 5 | 4 / 5 | 4 / 5 | 4 / 5 | 4 / 5 | 5 / 5 | 5 / 5 | 1 / 5 | 1 / 5|
>
> M-Mamba achieves performance gains in **58** out of 70 scenarios, with improvements becoming increasingly prominent at longer prediction horizons. This phenomenon is consistent with our architectural adjustments: removing the convolution operation and adopting a feature-independent matrix $\mathbf{A}$, which deemphasizes **local temporal dependencies**, allowing Mamba to better capture **long-term dependencies**. Recent studies, including iTransformer and ModernTCN, underscore the importance of a **global receptive field and long-term dependency** modeling for effective long-term time series forecasting. Specifically, iTransformer utilizes self-attention to achieve this, while ModernTCN leverages large convolution kernels. By removing modules emphasizing local dependencies, we enable Mamba to prioritize global dependency capture, as evidenced by the significant improvements in experimental results. These findings validate the effectiveness of our modifications in enhancing Mamba's capacity for long-term forecasting.
>
> - **Q1: Why MLP is optimal?**
>
> Thanks for the valuable question. We think MLP is optimal for its efficiency and potential to capture both correlations and irrelevances. For instance, in the extremely special case where there is no relationship between channels, MLP can be transformed into an identity module through learning. The universal approximation theorem also illustrates the potential of MLP. However, the original MLP cannot achieve good generalization effects because it is susceptible to distribution shifts. To cope with this, we propose GDD-MLP and Channel Mixup, which greatly enhances the generalization ability of MLP, as shown in Figure 4 in our paper. What's more, the recent NeurIPS 24 paper SOFTS[1], which proposes a centralized MLP architecture, demonstrates the strong ability of MLP in capturing cross-channel dependencies. The following table compares SOFTS with our CMamba, which shows that our model still outperforms it.
>
> ||ETTm1||ETTm2||ETTh1||ETTh2||Weather||Electricity||Traffic||
> |:-:|:-:|:-:|:-:|:-:|:-:|:-:|:-:|:-:|:-:|:-:|:-:|:-:|:-:|:-:|
> ||MSE|MAE|MSE|MAE|MSE|MAE|MSE|MAE|MSE|MAE|MSE|MAE|MSE|MAE|
> |SOFTS|0.393|0.403|0.287|0.330|0.449|0.442|0.373|0.400|0.255|0.278|0.174|0.264|**0.409**|0.267|
> |CMamba|**0.376**|**0.379**|**0.273**|**0.316**|**0.433**|**0.425**|**0.368**|**0.391**|**0.237**|**0.259**|**0.169**|**0.258**|0.444|**0.265**|
>
> [1] Han, Lu, et al. "SOFTS: Efficient Multivariate Time Series Forecasting with Series-Core Fusion." NeurIPS 2024.

---

> > ### Comment · Reviewer_ah3a · 2024-11-25
> >
> > I mentioned at the end of the 5th point of the weaknesses ‘In Time-SSM Table 2 gives a detailed comparison with the patch Mamba (Mamba4TS).’， that it doesn't make much sense to compare it with the vanilla  Mamba model. What I can see is that your improvement is based on the existing Patching, but it doesn't have an obvious effect on enhancing the existing methods. Therefore, the overall contribution is insufficient. The cross-channel approach is also a conventional idea that lacks innovation.

---

> ### Author Response · Authors · 2024-11-25
>
> It seems that you misunderstood our experimental settings. We highlight in our response to W5 that: **we only modify the backbone, so GDD-MLP and Channel Mixup are still adopted**, which means that the only difference between Mamba and M-Mamba is our modification, that is, **Mamba here also takes patch-wise inputs (i.e., it is PatchMamba)**.
>
> In terms of the cross-channel approach, we have proved in **Fig. 4** that the vanilla MLP does not work well. Our approach, GDD-MLP and channel mixup, though simple, does enable the MLP architecture to achieve good results in capturing cross-channel dependencies.

---

> > ### Comment · Reviewer_ah3a · 2024-11-27
> >
> > It seems that you don't understand the question I've repeated twice. Or is it that you don't understand the basic logic of the ablation experiment? I can't see the difference in the performance between the M-mamba and PatchMamba models. What does this ablation experiment have to do with the GDD-MLP? Or are you just unwilling to admit this problem?

---

### Official Review · Reviewer_Z3ph · 2024-10-30

**Soundness:** 2
**Presentation:** 3
**Contribution:** 3
**Rating:** 5
**Confidence:** 4

**Summary:**

In this work, the authors proposed model, CMamba, incorporates a modified Mamba (M-Mamba) module for temporal dependencies modeling, a global data dependent MLP (GDD-MLP) to effectively capture cross-channel dependencies, and a Channel Mix-up mechanism to mitigate over-fitting.

**Strengths:**

1.  The authors tailor the vanilla Mamba module for multivariate time series forecasting. A modified Mamba (M-Mamba) module is proposed for better cross-time dependencies modeling.
 2. The authors enable Mamba to capture multivariate correlations using the proposed global data dependent MLP (GDD-MLP) and Channel Mix-up. Coupled with the M-Mamba module, the proposed CMamba captures both cross-time and cross-channel dependencies to learn better representations for multivariate time series forecasting.

**Weaknesses:**

1. The description of the contribution seems to be incremental innovation, and the modeling of dependencies across time variables is the purpose, rather than a clear design motivation. In addition, different exogenous time series have positive or negative effects on the target series. How should this dependence be measured?
2. It can be seen from the experimental results in Table 1 that the performance improvement is weak. For example, on the ETTm1 dataset, the authors' method improved the MSE metrics by about 2% compared to the SOTA method (i.e., TimeMixer). If the authors claim that the cross-time dependence should produce significant performance improvement, the current experimental results are contrary to the intuition. After all, the dependencies between multiple time variables in electric time series data are much clearer.
3. Especially in the ablation experiments of Table 2 with standard Mamba, we can find that the performance improvement is close to the test error. In terms of the MSE metric, the standard Mamba is 0.240 and the proposed method is 0.237, which is only 1.25% improvement.
4. The reviewer is also concerned about the extent to which such modifications impose computational overhead on the several datasets. Especially the GDD-MLP module and CMamba-encoder module, is it because these complex components increase the complexity of the proposed prediction model and thus improve the prediction effect? Therefore, authors are expected to provide training and inference time on multiple time series datasets.

**Questions:**

Please see weaknesses!

---

> ### Author Response · Authors · 2024-11-16
> **Response to Reviewer Z3ph (Part 1)**
>
> We sincerely thank Reviewer Z3ph for providing thorough and insightful comments. Here are our responses to your concerns and questions.
>
> - **W1 a: Unclear description of contributions and motivation.**
>
> Thanks for your advice. The modeling of dependencies across time variables is both our motivation and purpose. Our description follows the following pipeline:
> 1. There are relationships between different time variables (shown in **Fig. 1**) but Mamba does not model cross-variate dependencies (Motivation )
> 2. --> We want to design a module to model cross-variate dependencies (Purpose) --> How?
> 3. --> The success of iTransformer: data-dependent and global receptive field (Motivation)
> 4. --> GDD-MLP and channel mixup. We hope that the above pipeline is more clear and solves your concerns.
>
> - **W1 b: The model does not consider the influence of exogenous variables.**
>
> Thanks for your advice. Nowadays, there are two mainstream directions in multivariate time series prediction:
> 1. predicting by better modeling the relationship (cross-time and cross-channel) between endogenous variables;
> 2. introducing exogenous variables to assist prediction.
>
> In this paper, we mainly focus on the first direction. Both Table 1 and Table 3 have shown the effectiveness of CMamba in modeling these dependencies. We believe that exogenous variables are important and useful for time series prediction. **We can introduce them into the model through attention, concat, or even feature addition**. However, as we said in **Appendix C.6 Limitations**, this will be a direction we can explore in the future. In addition, in some real-world scenarios, it is much more **difficult to obtain exogenous variables than endogenous variables**. Therefore, we believe that how to rely solely on endogenous variables to perform time series forecasting is a more universal topic, and the corresponding results can also be well applied to scenarios that consider exogenous variables.
>
> - **W2: The improvement is not significant after considering cross-channel dependencies.**
>
> Thanks for the comments. If I understand correctly, you are referring to cross-channel (variable) dependencies rather than cross-time dependencies. The following descriptions are provided from the perspective of cross-channel dependencies:
>
> Different models have different design focuses. For instance, TimeMixer focuses on cross-time dependency modeling and simply utilizes the channel-independent strategy. It achieves this with **season & trend decomposition and multiscale mixing**. However, our model does not employ these techniques. Hence, I think it is unfair to deny the necessity of cross-channel dependency modeling by comparing the improvement rate of CMamba relative to other models (like, TimeMixer). TimeMixer focuses more on cross-time modeling while we focus on cross-channel modeling. As shown in the following table, modeling cross-channel dependencies do improve CMamba's performance.
>
> ||ETTm1||ETTm2||Weather||Electricity||
> |:-:|:-:|:-:|:-:|:-:|:-:|:-:|:-:|:-:|
> ||MSE|MAE|MSE|MAE|MSE|MAE|MSE|MAE|
> |CMamba w / o cross-channel|0.387|0.387|0.282|0.322|0.255|0.271|0.190|0.268|
> |CMamba|**0.376**|**0.379**|**0.273**|**0.316**|**0.237**|**0.259**|**0.169**|**0.258**|
> |Improvement rate|2.87\%|2.16\%|3.07\%|1.94\%|6.95\%|4.14\%|10.95\%|3.89\%|
>
> Admittedly, different datasets have different improvement rates because the complexity of the variable relationships in different datasets differs. For instance, some variables in the ETT dataset have obvious proportional relationships (e.g., *MULL* (Middle UseLess Load) is roughly equivalent to half of *HULL* (High UseLess Load)). Such dependencies are easy to model, and even a linear layer can simulate this kind of relationship, i.e., the input is proportional, so the output after linear projection is naturally proportional. This is why Linear-based models can always achieve sota performance on the ETT dataset. In more complex datasets, such as Weather and Electricity, our method shows advantages.

---

> > ### Author Response · Authors · 2024-11-16
> > **Response to Reviewer Z3ph (Part 2)**
> >
> > - **W3: The improvement is not significant after turning Mamba to M-Mamba.**
> >
> > Thanks for the comments. All methods are run three times for robust performance, as shown in **Table 7 in Appendix A.2 Hyperparameters**, where we report the average performance with standard deviation. We have tried our best to avoid testing errors. Here, we exhibit the performance of our Mamba design and the vanilla Mamba under other datasets. The results show that our Mamba design does perform better. However, since we are **simplifying the original Mamba architecture** based on the fact that long-term dependencies are more important in long-term time series forecasting (so we remove the conv1d) and different time steps share similar characteristics (so we use 'FI' transition matrix **A**), I think this level of improvement is acceptable. More results and evaluation are presented in the newly added **Appendix C.1**
> >
> > ||ETTm1||ETTm2||ETTh1||ETTh2||Weather||Electricity||Traffic||
> > |:-:|:-:|:-:|:-:|:-:|:-:|:-:|:-:|:-:|:-:|:-:|:-:|:-:|:-:|:-:|
> > ||MSE|MAE|MSE|MAE|MSE|MAE|MSE|MAE|MSE|MAE|MSE|MAE|MSE|MAE|
> > |Vanilla Mamba|0.383|0.381|0.277|0.319|0.435|0.428|0.370|0.393|0.240|0.261|0.173|0.260|0.443|0.264|
> > |M-Mamba|**0.376**|**0.379**|**0.273**|**0.316**|**0.433**|**0.425**|**0.368**|**0.391**|**0.237**|**0.259**|**0.169**|**0.258**|0.444|0.265|
> >
> > - **W4: Lack of description of computational overhead.**
> >
> > Thanks for your suggestions. We have added the analysis of model efficiency in **Appendix C.2 Efficiency Analysis** in the newly updated version. Here we show the results on the Weather dataset, with more details presented in **Appendix C.2**.
> >
> > |Model|RLinear|TimeMixer|TimesNets|CMamba|ModernTCN|iTransformer|PatchTST|
> > |:-:|:-:|:-:|:-:|:-:|:-:|:-:|:-:|
> > |Parameters|69.84k|0.22M|1.25M|**1.35M**|3.09M|5.15M|10.74M|
> > |TrainingTime (ms/iter)|13|43|325|**29**|29|28|33|
> > |MSE|0.364|0.345|0.365|**0.339**|0.343|0.358|0.354|
> >
> > We can find that CMamba is the most lightweight model on average except for the Linear-based model (e.g., RLinear and TimeMixer). This indicates that CMamba’s excellent performance stems more from its capacity to capture historical time series patterns than from an increase in parameters.

---

> > > ### Comment · Reviewer_Z3ph · 2024-11-16
> > > **Feedback for authors**
> > >
> > > Dear Authors,
> > >
> > > Thank you for your rebuttal! However, as in the second part of your response, the removal of components from Mamba and a series of improvements to the performance point of view only marginally improved. Even if the error is excluded, the effect is relatively limited. Therefore, I maintain the previous score.

---

> ### Author Response · Authors · 2024-11-17
> **Response to Reviewer Z3ph**
>
> Dear Reviewer-Z3ph,
>
> We appreciate Reviewer-Z3ph's feedback and would like to take this opportunity to further clarify two key contributions of our work:
> - **Firstly, we simplify the vanilla Mamba block**. We propose a simplified Mamba block by removing the `conv1d` layer and simplifying matrix `A`, as these components were found to negatively impact long-term dependency modeling. The improvements achieved with the simplified design are particularly notable for longer prediction horizons, as illustrated in the table below. Given that **removing these components enhances model performance, albeit modestly, we advocate for the adoption of this streamlined version of Mamba, aligning with the principle of model simplicity and efficiency**.
>
> || | ETTm1 || ETTm2 || ETTh1 || ETTh2 || Electricity || Weather || Traffic||
> |:-:|:-:|:-:|:-:|:-:|:-:|:-:|:-:|:-:|:-:|:-:|:-:|:-:|:-:|:-:|:-:|
> || | MSE | MAE  | MSE | MAE | MSE | MAE | MSE | MAE | MSE | MAE | MSE | MAE | MSE | MAE|
> |Mamba| 96| 0.312 | 0.341 | 0.173 | 0.251 | 0.376 | 0.390 | 0.283 | 0.331 | 0.140 | 0.231 | 0.152 | 0.189 | 0.404 | 0.250|
> || 192 | 0.365 | 0.367 | 0.239 | 0.296 | 0.423 | 0.417 | 0.360 | 0.381 | 0.159 | 0.247 | 0.201 | 0.236 | 0.429 | 0.255|
> || 336 | 0.398 | 0.392 | 0.300 | 0.336 | 0.468 | 0.439 | 0.416 | 0.422 | 0.177 | 0.265 | 0.264 | 0.283 | 0.455 | 0.265|
> || 720 | 0.455 | 0.427 | 0.398 | 0.393 | 0.474 | 0.466 | 0.419 | 0.437 | 0.214 | 0.298 | 0.344 | 0.337 | 0.483 | 0.286|
> || Avg | 0.383 | 0.381 | 0.277 | 0.319 | 0.435 | 0.428 | 0.370 | 0.393 | 0.173 | 0.260  | 0.240 | 0.261 | 0.443 | 0.264|
> |M-Mamba| 96| **0.308** | **0.338** | **0.171** | **0.248** | **0.372** | **0.386** | **0.281** | **0.329** | 0.141 | 0.231 | **0.150** | **0.187** | 0.414 | 0.251|
> || 192 | **0.359** | **0.364** | **0.235** | **0.292** | **0.422** | **0.416** | 0.361 | 0.381 | **0.157** | **0.245** | **0.200** | **0.236** | 0.432 | 0.257|
> || 336 | **0.390** | **0.389** | **0.296** | **0.334** | **0.466** | **0.438** | **0.413** | **0.419** | **0.175** | **0.265** | **0.260** | **0.281** | **0.446** | **0.265**|
> || 720 | **0.447** | **0.425** | **0.392** | **0.391** | **0.470** | **0.461** | **0.419** | **0.435** | **0.203** | **0.289** | **0.339** | **0.334** | 0.485 | 0.286|
> || Avg | **0.376** | **0.379** | **0.273** | **0.316** | **0.433** | **0.425** | **0.368** | **0.391** | **0.169** | **0.258** | **0.237** | **0.259** | 0.444 | 0.265|
> ||Promotion Count | 5 / 5 | 5 / 5 | 5 / 5 | 5 / 5 | 5 / 5 | 5 / 5 | 4 / 5 | 4 / 5 | 4 / 5 | 4 / 5 | 5 / 5 | 5 / 5 | 1 / 5 | 1 / 5|
>
> - **Secondly and more importantly, we focus more on cross-channel dependency modeling**. The core innovation of our work lies in **capturing cross-channel dependencies effectively through the use of GDD-MLP and Channel Mixup**, which retain the MLP architecture and provide substantial improvements. Our experiments (refer to **Fig. 4** in the main text) demonstrate that neither vanilla MLP nor generic mixup techniques work as effectively. In the following table, we replace the Mamba backbone in our model with MLP. We name the resulting model CMLP. As illustrated below, **CMLP significantly outperforms vanilla MLP and even achieves comparable or superior results to iTransformer (and other benchmarks compared in our paper), with dramatically fewer parameters**. This underscores the versatility and efficiency of GDD-MLP and Channel Mixup.
>
> |||CMLP|||Pure MLP|||iTransformer|||
> |:-:|:-:|:-:|:-:|:-:|:-:|:-:|:-:|:-:|:-:|:-:|
> ||T|MSE|MAE|Parameters|MSE|MAE|Parameters|MSE|MAE|Parameters|
> |Weather|96|**0.151**|**0.190**|696k|0.206|0.176|694k|0.174|0.214|4.83M|
> ||192|**0.202**|**0.237**|991k|0.226|0.252|989k|0.221|0.254|4.88M|
> ||336|**0.258**|**0.280**|1.43M|0.276|0.290|1.43M|0.278|0.296|4.96M|
> ||720|**0.341**|**0.335**|2.61M|0.354|0.341|2.61M|0.358|0.349|5.15M|
> ||Avg|**0.238**|**0.260**|1.43M|0.258|0.274|1.43M|0.258|0.279|4.96M|
> |Electricity|96|**0.145**|**0.235**|1.00M|0.173|0.252|694k|0.148|0.240|4.83M|
> ||192|**0.162**|**0.250**|1.30M|0.180|0.260|989k|**0.162**|0.253|4.88M|
> ||336|**0.178**|**0.268**|1.74M|0.196|0.276|1.43M|**0.178**|0.269|4.96M|
> ||720|**0.218**|**0.301**|2.92M|0.242|0.314|2.61M|0.225|0.317|5.15M|
> ||Avg|**0.176**|**0.263**|1.74M|0.198|0.275|1.43M|0.178|0.270|4.96M|
>
> We hope this addresses your concerns and emphasizes the contributions of our work.

---

> > ### Comment · Reviewer_Z3ph · 2024-11-17
> > **Feedback for authors**
> >
> > Dear Authors,
> >
> > Many thanks to the authors for adding quite a lot of experiments in such a short rebuttal time. It is clear that removing the last conv1d layer is technically a simple operation and, crucially, not a significant increase in computational efficiency and performance.
> > The reviewers suggest that future authors could supplement theoretical analysis to explain the role of various Mamba components. In addition, it is worth considering whether it is reasonable to remove some potentially non-essential components of the Mamba structure.
> >
> > Best Regards,
> > Reviewer

---

### Official Review · Reviewer_86ct · 2024-10-31

**Soundness:** 3
**Presentation:** 2
**Contribution:** 2
**Rating:** 5
**Confidence:** 3

**Summary:**

CMamba is a new state-space model for multivariate time series prediction. It combines the Mamba model with global data-dependent MLP and channel mixing strategies to enhance capture across time series and cross-channel dependencies. Experiments on seven real-world data sets showed that CMamba achieved significant improvements in predictive performance.

**Strengths:**

The main advantage of this paper is that uses a Mamba-base model and channel-independent and mixed strategies to improve the performance of time series prediction.At the same time, the article is well written and clearly expressed

**Weaknesses:**

CMamba model mainly focuses on the prediction of endogenous variables of multivariable time series but does not involve the influence of exogenous variables. On large-scale channel data sets, such as weather and power data, the model has limited improvement in the MAE index, especially on traffic data sets. This may be due to the strong periodicity of traffic data covering up the physical connection between channels. Therefore, the inclusion of external variables and the use of prior knowledge of the relationships between channels may further improve the prediction accuracy.

**Questions:**

1. What are the specific advantages of the Mamba-based model compared to the MLP-based model?
2. Some of the baseline models MAE/MSE appear to differ from those reported in the original article(ModernTCN)

---

> ### Author Response · Authors · 2024-11-16
> **Response to Reviewer 86ct (Part 1)**
>
> We sincerely thank Reviewer 86ct for providing thorough and insightful comments. Here are our responses to your concerns and questions.
>
> - **W1: CMamba does not involve the influence of exogenous variables.**
>
> Thanks for pointing out the weakness. We have discussed the limitations of our model in **Appendix C.6 Limitations**, where both exogenous variables and limited MAE improvement on the Traffic dataset are discussed. In this work, **we mainly focus on cross-time and cross-channel dependencies modeling with endogenous variables, which is a well-built topic**[1][2]. Both Table 1 and Table 3 show the effectiveness of CMamba in modeling these dependencies.
>
> Admittedly, some recent work[3] has demonstrated that exogenous variables can improve the prediction accuracy of endogenous variables. However, exogenous variables are not the focus of this paper, so we only regard them as a direction for future exploration. In addition, in some real-world scenarios, it is much more **difficult to obtain exogenous variables than endogenous variables**. Therefore, we believe that how to rely solely on endogenous variables to perform time series forecasting is a more universal topic, and the corresponding results can also be well applied to scenarios that consider exogenous variables.
>
> [1] Liu, Yong, et al. "itransformer: Inverted transformers are effective for time series forecasting." ICLR 2024.
>
> [2] Nie, Yuqi, et al. "A time series is worth 64 words: Long-term forecasting with transformers." ICLR 2023.
>
> [3] Wang, Yuxuan, et al. "Timexer: Empowering transformers for time series forecasting with exogenous variables." NeurIPS 2024.
>
> - **Q1: What are the specific advantages of the Mamba-based model compared to the MLP-based model?**
>
> Thanks for the question. Unlike other fields, Linear/MLP-based models have been proven to be very powerful models in time series prediction[4][5]. The advantage of Mamba-based or Transformer-based models over Linear/MLP-based models is their data-dependent ability. This enables these models to cope with more diverse scenarios. For instance, Transformer-based models can adaptively adjust the weights of feature fusion through attention scores to cope with different data inputs, while the Linear/MLP-based models can only use constant weights. Mamba-based models also have characteristics similar to those of Transformer-based models.
>
> In the following table, we replace the M-Mamba backbone of CMamba with MLP. We name the resulting model **CMLP**. The results show that the Mamba backbone, with fewer parameters, performs better than the MLP backbone. What's more, CMLP largely outperforms pure MLP and iTransformer, which demonstrates the effectiveness of our modeling of cross-channel dependencies.
>
> |||CMamba|||CMLP|||Pure MLP|||iTransformer||
> |:-:|:-:|:-:|:-:|:-:|:-:|:-:|:-:|:-:|:-:|:-:|:-:|:-:|
> ||T|MSE|MAE|Parameters|MSE|MAE|Parameters|MSE|MAE|Parameters|MSE|MAE
> |Weather|96|**0.150**|**0.187**|387k|0.151|0.190|696k|0.206|0.176|694k|0.174|0.214|
> ||192|**0.200**|**0.236**|535k|0.202|0.237|991k|0.226|0.252|989k|0.221|0.254|
> ||336|0.260|**0.280**|756k|**0.258**|**0.280**|1.43M|0.276|0.290|1.43M|0.278|0.296|
> ||720|**0.339**|**0.334**|1.35M|0.341|0.335|2.61M|0.354|0.341|2.61M|0.358|0.349|
> ||Avg|**0.237**|**0.259**|756k|0.238|0.260|1.43M|0.258|0.274|1.43M|0.258|0.279|
> |Electricity|96|**0.141**|**0.231**|693k|0.145|0.235|1.00M|0.173|0.252|694k|0.148|0.240|
> ||192|**0.157**|**0.245**|841k|0.162|0.250|1.30M|0.180|0.260|989k|0.162|0.253|
> ||336|**0.175**|**0.265**|1.06M|0.178|0.268|1.74M|0.196|0.276|1.43M|0.178|0.269|
> ||720|**0.203**|**0.289**|1.65M|0.218|0.301|2.92M|0.242|0.314|2.61M|0.225|0.317|
> ||Avg|**0.169**|**0.258**|1.06M|0.176|0.263|1.74M|0.198|0.275|1.43M|0.178|0.270|
>
> [4] Zeng, Ailing, et al. "Are transformers effective for time series forecasting?." AAAI 2023.
>
> [5] Wang, Shiyu, et al. "Timemixer: Decomposable multiscale mixing for time series forecasting." ICLR 2024.

---

> ### Author Response · Authors · 2024-11-16
> **Response to Reviewer 86ct (Part 2)**
>
> - **Q2: MAE/MSE of some baseline models appear to differ from those reported in the original article (e.g., ModernTCN).**
>
> Thanks for your carefulness. Some results differ from those reported in the original article for two reasons:
> 1. These models use look-back windows of different lengths as input in the original paper. For instance, ModernTCN takes look-back length as a hyperparameter, and PatchTST adopts a look-back length of 336 or 512. Following the setting of most recent models, e.g., iTransformer, and TimesNet, we unify the look-back length as **96**.
> 2. The `drop_last` bug stemming from Informer. This bug comes from an incorrect implementation of the dataloader, where the test dataloader uses `drop_last=True`, which may exclude a significant portion of test data, particularly with large batch sizes, leading to unfair model comparisons. It has influenced many models, e.g., PatchTST, ModernTCN, and TimeMixer. We fix the bug in our implementation. Therefore, the MAE/MSE of the models affected by the bug is different from the original results.

---

> > ### Comment · Reviewer_86ct · 2024-11-25
> >
> > Now I am still curious about the feasibility of applying mamba as a backbone network to time series tasks. Since mamba network itself is still controversial, I keep the original score.

---

### Author Response · Authors · 2024-11-16
**Summary of Revisions and Rebuttals**

Dear reviewers:

We sincerely thank you for your thorough and insightful reviews. Your constructive feedback has been invaluable in helping us refine and improve our manuscript.

We are pleased that the majority of reviewers recognize the contributions of GDD-MLP and Channel Mixup, which enable MLPs to effectively capture cross-channel dependencies while demonstrating strong versatility and efficiency. Some reviewers raise concerns about the introduction and related works writing, efficiency analysis, and inadequate ablation experiments about Mamba. We have tried our best to address these concerns in the new updated version:

1. **Clearer motivation**: We have elaborated on the deficiencies in existing research, providing a more compelling motivation for our work (see **Lines 53-59**).
2. **Enhanced Description of Transformer**: To avoid ambiguity, we have clarified our description of Transformer components (see **Lines 65–69**).
3. **Removal of Experimental Results from the Introduction**: We have removed Fig.2 in the introduction section, which includes some experimental results.
4. **Expanded Related Works**: We have added more detailed descriptions on the current state of research in state space models for time series forecasting (see **Lines 127-139**).
5. **More comprehensive ablation studies**: We add **Appendix C.1**, where full ablation results about our Mamba design are presented.
6. **Efficiency analysis**: We have added an efficiency evaluation in **Appendix C.2** to provide a more thorough understanding of the model’s computational performance.

However, some reviewers still have concerns about the applicability of Mamba in the time series domain, and about our removal of some components of Mamba, even though our model has shown performance that exceeds the SOTA model in Table 1 of the main text and Table 9 of the appendix. Some reviewers hope that we can conduct a theoretical analysis of the role of each component in Mamba in time series prediction, but the workload is enough to form a new paper. We understand the reviewers' concerns. However, we also hope that the reviewers can consider the broader contributions of our paper beyond Mamba, particularly the GDD-MLP and channel mixup mechanisms. The results presented in the rebuttal underscore that even with a fully MLP-based backbone, incorporating GDD-MLP and Channel Mixup allows the model to achieve competitive results.

Once again, we extend our gratitude to the reviewers for their time and effort in evaluating our work. Your insights have been instrumental in enhancing the quality of our paper.

Best regards,

Authors

---

### Meta-Review · Area_Chair_AS6Q · 2024-12-19

**Metareview:**

This paper proposes a new model, called CMamba, for multivariate time series forecasting. Based on a modified Mamba (M-Mamba) module that aims to model temporal dependencies, a global data-dependent MLP (GDD-MLP) is incorporated to capture cross-channel dependencies and a Channel Mixup mechanism is incorporated to mitigate overfitting.

Major strengths:
- The effectiveness of Mamba-based models for multivariate time series forecasting is still an open problem that needs more study.
- The presentation is generally clear.

Major weaknesses:
- This paper is a bit lack of focus. It covers three aspects (as summarized above), each of which is not a very significant contribution with sufficient justification.
- For some large datasets, the improvement is quite limited.

We appreciate the effort of the authors for addressing the comments and concerns raised by the reviewers in their reviews. Additional experiments were also conducted to address some of the concerns. Despite the effort put into this work, the question of whether Mamba-based models are effective for multivariate time series forecasting remains unanswered. One may wonder whether other models can also benefit from the proposed techniques. Addressing these issues more thoroughly will make this paper more ready for publication.

**Additional Comments On Reviewer Discussion:**

The authors engaged in discussions with the reviewers and provided additional experiment results. However, most reviewers still hold a negative view and the only one who is positive about the paper only thinks that it is “marginally above the acceptance threshold” but is unwilling to cast a stronger vote.

---

### Decision · Program_Chairs · 2025-01-22

Reject